# Mid-Pliocene not analogous to high $CO_2$ climate when considering Northern Hemisphere winter variability

Arthur M. Oldeman[1], Michiel L.J. Baatsen[1], Anna S. von der Heydt[1,2], Aarnout J. van Delden[1], and Henk A. Dijkstra[1,2]

[1]Institute for Marine and Atmospheric research Utrecht (IMAU), Department of Physics, Utrecht University, 3584 CC Utrecht, the Netherlands
[2]Centre for Complex Systems Studies, Utrecht University, 3584 CE Utrecht, the Netherlands

**Correspondence:** Arthur Oldeman (a.m.oldeman@uu.nl)

**Abstract.** In this study, we address the question of whether the mid-Pliocene climate can act as an analogue for a future warm climate with elevated $CO_2$ concentrations, specifically regarding Northern Hemisphere winter variability. We use a set of sensitivity simulations with the global coupled climate model CESM1.0.5 (CCSM4-Utr), that is part of the PlioMIP2 model ensemble, to separate the response to a $CO_2$ doubling and to mid-Pliocene boundary conditions other than $CO_2$. In the $CO_2$ doubling simulation, the Aleutian low deepens, and the Pacific-North American pattern (PNA) strengthens. In response to the mid-Pliocene boundary conditions, sea-level pressure variance decreases over the North Pacific, the PNA becomes weaker, and the North Pacific Oscillation (NPO) becomes the dominant mode of variability. The mid-Pliocene simulation shows a weak North Pacific jet stream that is less variable in intensity, but has a high level of variation in jet latitude, consistent with a dominant NPO, and indicating that North Pacific atmospheric dynamics become more North Atlantic-like. We demonstrate that the weakening of the Aleutian low, and subsequent relative dominance of the NPO over the PNA, is related to shifts in tropical Pacific convection. Variability in the North Atlantic shows little variation between all simulations. The opposite response in North Pacific winter variability to elevated $CO_2$ or mid-Pliocene boundary conditions demonstrate that the mid-Pliocene climate cannot serve as a future analogue in this regard.

## 1 Introduction

Our present climate is warming due to humans increasing the concentration of greenhouse gases in the atmosphere, and making accurate projections of our future climate is an important necessity. Future climate is dependent on the emission pathway we choose, and determined by the response of the climate system itself to increased $CO_2$, through feedbacks and natural variations. One way of investigating the response of the climate system to warm conditions, is by looking at the past. We can use equilibrium climate model simulations to investigate the climate response to elevated levels of atmospheric $CO_2$. The geological past was in equilibrium with forcing on climatological timescales, and saw multiple periods with elevated atmospheric $CO_2$ and temperatures of which there is a reasonable amount of geological evidence and reconstructions (see also Chen et al., 2021, FAQ1.3). The most recent period with similar $CO_2$ concentrations as the present-day was the mid-Pliocene warm period, approximately 3 million years (Ma) ago (Tierney et al., 2020). It had a similar geography - compared

to earlier geological warm periods - to the present-day. The most notable differences include a reduction of the Greenland and West Antarctic ice sheets as well as a closure of the Bering Strait and Canadian Arctic Archipelago. It has been called the 'best analogue' for near-future climate, because surface temperatures at the end of this century would be most similar to mid-Pliocene temperatures, both following the RCP4.5 and RCP8.5 scenarios, compared to other past climate periods (Burke et al., 2018).

The mid-Pliocene has been investigated using coupled climate models as part of two phases of the Pliocene Model Intercomparison Project (PlioMIP). The most recent PlioMIP2 employs a time-slice approach in order to compare model simulations and proxy reconstructions in a detailed way. Several model-data comparisons and ensemble investigations have been done in the PlioMIP2, for example concerning general mean climate (Haywood et al., 2020), the global hydrological cycle (Han et al., 2021), Arctic warming (De Nooijer et al., 2020), the Atlantic Meridional Overturning Circulation (AMOC, Zhang et al., 2020; Weiffenbach et al., 2023) and El Niño-Southern Oscillation (ENSO, Oldeman et al., 2021; Pontes et al., 2022). Some of these studies show very different climate responses to the forcing changes in the mid-Pliocene compared to near-future projections, for example a stronger AMOC (Weiffenbach et al., 2023) compared to a weakening of the AMOC in the projected future, as well as strongly reduced ENSO variability (Oldeman et al., 2021), which is not projected for near-future climate.

The equilibrated mid-Pliocene climate response might be more suited to compare with long-term future climate projections than near-future projections which are still in a transient state. For example, the mid-Pliocene saw atmospheric $CO_2$ concentration of 400 ppm, which is similar to the equilibrated levels of the SSP1-2.6 scenario in the year 2300 (Chen et al., 2021). Also, Haywood et al. (2020) find a PlioMIP2 mean surface temperature increase of 3.2°C, which is similar to the global average warming of the RCP4.5 scenario in the year 2400 (Lyon et al., 2022). On even longer timescales, Feng et al. (2022) argue that the mid-Pliocene changes in ice sheets and vegetation are a long-term Earth system feedback to elevated atmospheric $CO_2$.

But some Pliocene boundary conditions will not be shared in a future climate, including the closure of the Bering Strait and the Canadian Arctic Archipelago. So, when considering the mid-Pliocene as a future analogue, we should consider the climate response or sensitivity to each boundary condition. Burton et al. (2023) use a linear fractionation method to show that $CO_2$ is the most important forcing for global surface temperatures and precipitation in the mid-Pliocene, employing a set of sensitivity studies of a selection of PlioMIP2 model contributions. However, especially at high latitudes surface temperatures respond more strongly to boundary conditions other than $CO_2$. This agrees with earlier work by Otto-Bliesner et al. (2017) as well as Chandan and Peltier (2018), that show large sensitivity of the northern high latitude mid-Pliocene climate to closed Arctic gateways, which induce a larger surface temperature and sea-ice response than the reduced Greenland ice sheet, or elevated atmospheric $CO_2$.

Apart from studying mean climate response in a warm past climate, it is useful to study the response of climate variability. Since contemporary climate change is transient, it is hard to assess changes to modes of variability that operate on interannual and decadal timescales. Investigating climate variability in equilibrated simulations can help to separate internal variability change to the forced response from transient trends. Within the PlioMIP2, most attention to climate variability has focused on ENSO (Oldeman et al., 2021; Pontes et al., 2022), which was found to be reduced in amplitude in a large majority of the models, consistent with findings from the earlier PlioMIP1 ensemble (Brierley, 2015). Earlier work by Hill et al. (2011) shows

reduced Northern Annular Mode (NAM) variability in the Pliocene, which was primarily attributed to the lowering of the Rocky Mountains. Both PlioMIP1 and PlioMIP2 employed a timeslice where the Rocky Mountain uplift had already occurred.

A feature where future climate projections fail to give a consistent response to increasing $CO_2$ levels is atmospheric variability in the Northern Hemisphere winter, specifically the Arctic Oscillation (AO; also referred to as the NAM) and its regional expression, the North Atlantic Oscillation (NAO) (Eyring et al., 2021). These modes represent an oscillation of mass between the subpolar latitudes and subtropics in the Northern Hemisphere, and are the leading modes of sea-level pressure (SLP) variability in the northern extratropics (Ambaum et al., 2001; Hurrell and Deser, 2010). Apart from the NAO, a less pronounced mode of SLP variability operates in the North Atlantic, namely the East Atlantic (EA) pattern, which usually has its center of action over the British isles (Barnston and Livezey, 1987). The NAO plays a significant role in North Atlantic and European climate, for example in extratropical storm tracks, and affects projections of temperature and precipitation on an interannual to decadal time scale (Deser et al., 2017; Iles and Hegerl, 2017). Uncertainties in climate projections over the Euro-Atlantic sector are largely related to atmospheric circulation variability (Woollings, 2010; Shepherd, 2014; Fereday et al., 2018), in particular because there is no consensus on how the AO and NAO respond to increasing $CO_2$ concentrations (Eyring et al., 2021). This is in part due to the large amplitude of internal variability related to the NAO compared to the amplitude of global warming (Osborn, 2004), as well as a considerable model spread in climate change simulations (Eyring et al., 2021). Though climate models are considered skilful in simulating the spatial features and variance of the present-day and historical NAO (Eyring et al., 2021), CMIP5 and CMIP6 models underestimate NAO variability and North Atlantic jet stream variations on a multi-decadal to centennial time-scale (Blackport and Fyfe, 2022).

The two leading modes of atmospheric winter variability over the North Pacific are the Pacific North-American (PNA) pattern and the North Pacific Oscillation (NPO) or West Pacific teleconnection (Barnston and Livezey, 1987; Linkin and Nigam, 2008). The PNA essentially captures most of the variability of the Aleutian low pressure system, and has a large influence on present-day North Pacific and North American winter climate. The NPO is strongly linked to latitudinal displacement of the Asian-Pacific jet and Pacific storm track variability, as well as the position and strength of the Aleutian low. Linkin and Nigam (2008) state that the NPO shows a strong connection to Arctic sea ice variability, as well as variations of the North Pacific jet stream latitude. North Pacific atmospheric variability is strongly related to atmospheric Rossby wave activity (Mo and Livezey, 1986), that can be excited through tropical convection associated with ENSO variability (Hoskins and Karoly, 1981). Chen et al. (2018) investigate a large ensemble of CMIP5 simulations and find a stronger PNA under high $CO_2$ forcing, but no consensus on the NPO. They find that CMIP5 models reasonably simulate spatial patterns and interannual variations of the present-day and historical PNA and NPO, but lack the capability to reproduce variations on a decadal timescale.

This brings us to the research question addressed in this study. Can the mid-Pliocene climate be used to determine the response of Northern Hemisphere (NH) winter atmospheric variability, such as the NAO, NAM and PNA, to increased $CO_2$? To answer this question, we will aim to answer two subquestions. 1. Is there a difference in the pre-industrial climate response to elevated $CO_2$, and to mid-Pliocene boundary conditions other than $CO_2$, including closed Arctic gateways and reduced ice sheets? 2. How do changes in mean winter climate relate to changes in atmospheric variability in the NH?

We will investigate NH winter variability in pre-industrial and mid-Pliocene simulations within one global climate model, the Community Earth System Model (CESM) version 1.0.5, that is a part of the PlioMIP2 ensemble. Since the equilibrated mid-Pliocene climate saw increased $CO_2$ levels as well as different boundary conditions such as topography, land-ice coverage and vegetation, we will use sensitivity simulations to identify the response to both boundary conditions. Starting from a pre-industrial reference simulation with present-day geography and 280 ppm atmospheric $CO_2$, we will assess the response to a $CO_2$ doubling (560 ppm) with present-day boundary conditions, as well as the response to mid-Pliocene boundary conditions at pre-industrial $CO_2$ level (280 ppm). Both simulations represent approximately 3°C warmer worlds, but due to different forcings (Baatsen et al., 2022).

Based on earlier research, we can hypothesize what we expect the answers to our research question will be. Menemenlis et al. (2021) investigate extratropical hydroclimate changes in the mid-Pliocene using a version of CCSM4, and find large precipitation changes linked to dynamical shifts in atmospheric rivers. A wintertime stationary wave train in the pre-industrial largely disappears in the mid-Pliocene, resulting in fewer atmospheric rivers over the eastern North Pacific region. They use sensitivity studies to show that this mean winter climate response is mainly caused by the changes in ice sheets and closed gateways, and not by an increased level of atmospheric $CO_2$. However, they do not study variability. Oldeman et al. (2021) show that ENSO variability in the PlioMIP2 ensemble is reduced, with the largest reduction in the model we employ. This is found to be related to a shift in the mean Pacific ITCZ position (Pontes et al., 2022). Since ENSO variability influences the NH atmospheric circulation, including atmospheric modes of variability such as the PNA and the NAO, and oceanic modes such as the PDO (see e.g. Mo and Livezey, 1986; Yeh et al., 2018; Domeisen et al., 2019), we can expect changes to NH atmospheric variability when we know that ENSO variability is reduced.

In the following section, we explain the model and simulations used, outline our analysis methods, and compare our pre-industrial simulations to reanalysis data. In section 3, we present results of mean winter climate as well as SLP winter variability for all sensitivity simulations. We also investigate jet stream variations related to SLP variability in response to the mid-Pliocene boundary conditions, and demonstrate how the tropical Pacific mean state is linked to changes in winter variability. Section 4 presents a physical interpretation and discussion of the results, as well as our answer to the question of the mid-Pliocene as a future analogue. A summary and outlook concludes the paper.

## 2 Methods and data

### 2.1 Model and simulations

#### 2.1.1 Model design and configuration

The model used in this study is one out of seventeen models in model ensemble PlioMIP2. The PlioMIP2 methodology for participating modelling groups is outlined in Haywood et al. (2016). In comparison to PlioMIP1, an enhanced set of boundary conditions is supplied (the PRISM4 reconstruction, see Dowsett et al. (2016)). The experimental setup is such that it represents a specific time-slice in the mid-Pliocene, the KM5c interglacial (3.205 Ma), where the orbital configuration was similar as

the present day. The provided mid-Pliocene boundary conditions include mid-Pliocene topography and bathymetry, coastlines, land surface properties (i.e. vegetation, soil type, and ice sheet coverage), as well as atmospheric composition. Some important aspects of the mid-Pliocene model geography compared to pre-industrial reference are the closure of the Bering Strait and the Canadian Arctic Archipelago (or Northwest Passage). This makes the Arctic Ocean isolated from the North Pacific ocean, as well as from the Labrador Sea. The Greenland Ice Sheet (GIS) is substantially reduced in spatial coverage, covering only part of southeastern Greenland and being reduced in height, affecting topography as well.

CESM is a fully coupled ocean-atmosphere-land-ice general circulation model. For specific use in paleoclimate modelling, version 1.0.5 is a suitable trade-off between model complexity and computational cost. The model version used here employs the atmosphere module CAM4, which is part of the Community Climate System Model version 4 (CCSM4), and can be considered a CMIP5 generation model. Within the PlioMIP2 ensemble our model is referred to as CCSM4-Utr. The atmospheric grid has a horizontal resolution of $\sim 2°$ ($2.5° \times 1.9°$ or $144 \times 96$ grid cells) and 26 vertical levels. Details on the model version, simulations, spin-up and general climate features can be found in Baatsen et al. (2022).

The PlioMIP2 model contributions vary in model complexity, resolution, implementation of the provided boundary conditions, as well as in the set of simulations performed. Each model has performed two core experiments, namely a pre-industrial reference simulation at 280 ppm $CO_2$ and a mid-Pliocene simulation at 400 ppm $CO_2$. Following the PlioMIP2 naming convention, these simulations are referred to $E^{280}$ and $Eoi^{400}$, respectively (the "o" and "i" referring to the implementation of Pliocene orography and ice sheets, and the number referring to the atmospheric $CO_2$ concentration). The CCSM4-Utr has performed an additional set of sensitivity simulations, employing present-day as well as mid-Pliocene geography, vegetation and ice sheets at different levels of atmospheric $CO_2$.

### 2.1.2 Performance within PlioMIP2 ensemble

Within the PlioMIP2, results of the pre-industrial $E^{280}$ and mid-Pliocene $Eoi^{400}$ simulations have been compared, both to proxy reconstructions, as well as to other ensemble members. Haywood et al. (2020) show that CCSM4-Utr is one of the best performing models in the PlioMIP2 regarding data-model comparison for each site. De Nooijer et al. (2020) study Arctic warming in the PlioMIP2 ensemble, and find that CCSM4-Utr is one of the best performing models considering the data-model comparison of Arctic temperature anomalies. CCSM4-Utr winter sea-ice extent also matches well with reconstructions, although seasonal sea-ice extent reconstructions are very limited. Tindall et al. (2022) show that a significant model-data bias arises in winter months, due to a potential warm bias in the data and a potential cold bias in the models, but this holds for all PlioMIP2 models.

The CCSM4-Utr mid-Pliocene $Eoi^{400}$ simulations have been included in several PlioMIP2 ensemble studies. Oldeman et al. (2021) show that the amplitude of mid-Pliocene ENSO variability was reduced compared to the pre-industrial, and that CCSM4-Utr has the most reduced ENSO variability of the ensemble (67% reduction compared to 24% in the ensemble mean). Pontes et al. (2022) also study PlioMIP2 ENSO and find a relation between the reduced ENSO and a northward shift in Pacific ITCZ. They show that CCSM4-Utr exhibits the largest northward shift of the ITCZ from October to February. Zhang et al. (2020) and Weiffenbach et al. (2023) show an increased AMOC strength in the PlioMIP2 ensemble. Weiffenbach et al. (2023)

furthermore explain that models with a closed Bering strait and Canadian Arctic Archipelago (including CCSM4-Utr) show reduced freshwater transport from the Arctic Ocean into the North Atlantic, causing an increase in subpolar North Atlantic salinity, which drives the stronger AMOC. In this case, CCSM4-Utr simulates a stronger AMOC in the mid-Pliocene with values close to the ensemble mean (22 Sv over 19 Sv in the pre-industrial).

### 2.1.3 Simulation specifics

The PlioMIP2 naming convention is used for our set of sensitivity simulations. The reference simulation is the pre-industrial $E^{280}$ with present-day boundary conditions (BCs; orography, ice sheets and vegetation) and 280 ppm atmospheric $CO_2$. We perform a $CO_2$ doubling simulation at present-day geography ($E^{560}$) to investigate the effect of radiative forcing alone. We also perform a mid-Pliocene BCs simulation at pre-industrial carbon levels ($Eoi^{280}$), to investigate the response to the mid-Pliocene orography, ice sheets and vegetation, but not to $CO_2$. Additionally, we perform a mid-Pliocene BCs simulation at 560 ppm $CO_2$ ($Eoi^{560}$) in order to investigate nonlinearity in the responses to the forcings. All simulations have spin-up times of around 3,000 model years, and analysis of the results of the $Eoi^{400}$ simulation is found in Baatsen et al. (2022).

For the sake of consistency, we use pre-industrial and $CO_2$ doubling simulations that employ a 'paleoclimate' vertical ocean diffusivity, that we also use in the mid-Pliocene simulations. The difference between this adapted $E^{280}$ and the $E^{280}$ that is used in most PlioMIP2 studies, as well as the motivation for using a different parametrisation, is discussed in Baatsen et al. (2022). The effect of this parametrisation difference was found to be very minimal on ocean surface temperatures and negligible for atmospheric variables. The $CO_2$ doubling simulation was also rerun with this paleo mixing parametrisation. The $E^{280}$ and $E^{560}$ simulations are spun-up from equilibrated simulations and have spin-up times of 4,000 and 2,500 model years, respectively (details on the spin up can be found in Supplementary Material Figure S1).

## 2.2 Data analysis and methodology

In order to investigate NH winter climate and variability, we study 200 years per simulation, and we use monthly mean Januaries only. Results for December-January-February (DJF) or climatological winter were found to give very similar results for the sake of this study. The vertical levels of the model are on a hybrid sigma scheme, so results in the vertical have been interpolated onto pressure levels.

We investigate the mean state, i.e. the mean of all 200 Januaries, as well as variability on top of this mean state. We consider SLP, near-surface air temperature (SAT) and precipitation, as well as zonal winds and potential vorticity (PV) in the higher troposphere. To identify the subtropical jet stream we investigate zonal wind at 200 hPa (U200). We also compute the location of the dynamical tropopause, here defined as the isoline of 2 PVU (Potential Vorticity Units) at 200 hPa. We also compute the dynamical tropopause defined as the largest zonal isobaric PV gradient per longitude, but it gave very similar results as using the 2PVU isoline. The dynamical tropopause at 200 hPa generally follows the subtropical jet stream core (Kunz et al., 2011).

To study winter climate variability we focus on the temporal behaviour of SLP. Before analysis, a Lowess smoothing using a 50 year moving window is removed from anomalies at each spatial grid point (alpha of 0.25). A window size of 50 years was chosen since we are mainly interested in interannual and decadal winter variability. Furthermore, CMIP5 models are

known to underestimate variations of multidecadal periods of the winter NAO and jet stream (Blackport and Fyfe, 2022). We use Empirical Orthogonal Function (EOF) analysis to separate the SLP variance in orthogonal modes of spatial and temporal variability. Spatial EOF patterns as well as their corresponding Principal Component (PC) time series are obtained from the Lowess filtered SLP anomalies and using a spatial weighting that corresponds to the cosine of the latitude. The EOF patterns are normalised using their spatial standard deviation.

We perform EOF analysis over the NH, as well as over the North Atlantic (NA, 90°W - 30°E) and North Pacific (NP, 120E° - 120°W) basins. The regions were chosen such that they include the centers of action of known present-day modes of climate variability, such as the AO, NAO and PNA, but also to be equal in longitudinal width. All EOF regions, including the NH, are bound in latitude between 20°N and 85°N. In the NH, we only consider the first EOF, which we call the AO (we prefer AO over NAM since the AO is often defined as SLP EOF, whereas the NAM is often defined as index of SLP anomalies). In the NA and NP regions, we consider the first and second EOF. Based on whether the spatial pattern represents a dipole or a monopole, we categorize the EOFs according their known nomenclature in the present-day; the NAO (NA dipole) and EA (NA monopole), and the NPO (NP dipole) and PNA (NP monopole). The dipole/monopole characterization is done by considering the shape and amplitudes of the peaks of the zonally averaged EOF patterns. While these modes may have a different pattern or prominence in a past climate with different boundary conditions, compared to the present day, we decided to use known nomenclature, both to avoid confusion and to follow past studies of paleoclimate variability (e.g. the NAO in the last glacial maximum; Rivière et al., 2010). The EOFs and PCs of the AO, NAO and NPO modes are defined such that the mean EOF amplitude in the region north of 75°N is positive, while the EA and PNA modes are defined such that the mean EOF amplitude in the region 50-60°N is negative. We also consider the percentage of variance explained by each EOF. We perform a variance bootstrapping method to quantify whether the variance explained per mode between two simulations has a statistically significant difference, which is defined as no overlap between the top 95% and bottom 5% (or vice versa) variance explained percentages obtained through the bootstrapping method.

We investigate in what way the different modes of winter variability in the whole NH and over the NP and NA basins are connected. We quantify the level of correlation by means of the Pearson correlation coefficient between the PCs of the corresponding modes. The correlation coefficient between the first and second mode in either the NP or NA will always be zero, since these EOFs are orthogonal by definition. We consider the correlation to be statistically significant when $p<0.05$.

We also perform an analysis of jet variations in the NP. For variations in the central NP jet stream, we limit ourselves to the 200 hPa level and the zonal mean between 160°E and 220°E (or 140°W). We define jet intensity as the maximum of the zonal wind velocity in this sector, and jet latitude as the latitude corresponding to this maximum. Furthermore, we investigate the jet stream in the phases of NPO mode, where we define the NPO+ phase as the average of the highest 5% (or ten Januaries) NPO PC values, and the NPO- phase as the average of the lowest 5% NPO PC values.

Lastly, we investigate the relationship between ENSO variability, tropical convection and the variability of SLP and the jet stream in the NP. We define ENSO variability through the Nino3.4 index, which is a measure of sea-surface temperatures (SST) anomalies in the Nino3.4 region in the equatorial Pacific (170°W - 120°W, 5°S - 5°N). We characterize tropical convection

using area-mean precipitation in the West equatorial Pacific (WEP, defined as 120°E - 180°E, 6°S - 6°N). We investigate the links between variables using correlations coefficients and regression (linear slope) between variables in time.

## 2.3 Validation of $E^{280}$ with reanalysis data

We compare the pre-industrial $E^{280}$ simulation to 19th and 20th century reanalysis data. It is not a one-on-one comparison,
since our pre-industrial simulation is an equilibrium scenario, while any kind of global reanalysis dataset will be influenced by human-induced climate change trends, and will thus be transient. We use monthly mean sea-level pressure data from the NOAA/CIRES/DOE 20th Century Reanalysis version 3 (from here on abbreviated as CR20). The reanalysis covers the period from 1836 to 2015 and assimilates surface pressure observations in combination with a forecasting model to estimate a set of atmospheric variables. An evaluation of the performance of the CR20v3 can be found in Slivinski et al. (2021). The length of
the CR20 dataset and the fact that it covers a period closer to what can be considered pre-industrial motivated us to use this reanalysis data over more recent reanalyses such as ERA5. The CR20 data is presented on a 1.0° latitude x 1.0° longitude grid, and we interpolate the data onto the model grid when computing differences between the $E^{280}$ and CR20. The same Lowess filtering (as with the simulation data) is applied to remove any trend due to anthropogenic climate change.

A comparison of mean SLP and the related patterns of variability between our pre-industrial simulation and reanalysis is
240 shown in Figure 1. Mean SLP (MSLP) is shown in Figure 1a. The root mean squared error (RMSE) of MSLP is 5.3 hPa. Differences with the CR20 include an overestimation of the subtropical high pressure regions, specifically over the North Atlantic and North-African regions, Central Asia and to a smaller extent over North America (up to +12.8 hPa). A lower MSLP is simulated in the subpolar low pressure region in the North Atlantic (up to -12.4 hPa), and the pattern extends more eastward. The North Pacific subpolar low pressure area, the Aleutian low, is well captured in spatial extent and amplitude.
Figure 1b shows SLP standard deviation (SD). SLP SD is generally overestimated in the $E^{280}$ (mean SLP SD difference is +1.6 hPa, RMSE of 2.3 hPa), especially over the North Pacific and Siberian Arctic (up to +5.5 hPa). SLP SD in the North Atlantic storm track region as well as to the east of the tip of Greenland is slightly underestimated. A higher (lower) modelled SLP variance in the North Pacific (North Atlantic) in comparison to observations is a known bias for most CMIP5 and CMIP6 models (Eyring et al., 2021).
Figures 1c-g show the spatial EOF patterns of SLP in the NH, NA and NP, including the percentage of variance explained for each mode. The NH mode, the AO, in Figure 1c explains more variance in the $E^{280}$ than in the reanalysis (28.7% over 22.2%), which is a statistically significant difference. The amplitudes of the North Atlantic centers of action are underestimated in the $E^{280}$, which can be expected since the $E^{280}$ simulated less SLP variance in that region.

The leading mode in the NP is the PNA (Figure 1f), explaining around 42% of SLP variance in both $E^{280}$ and CR20. The
255 second leading mode in the NP is the NPO (Figure 1d), explaining 25% (22%) of SLP variance in the $E^{280}$ (CR20). The $E^{280}$ simulates both modes well in terms of spatial pattern, although the amplitude of the centers of action are a bit less pronounced and the variance more spread out over the region. The percentages of variance explained are not statistically different.

The leading mode in the NA is the NAO (Figure 1e), explaining 32% (33%) of SLP variance in the $E^{280}$ (CR20). The amplitude of the southern node is slightly underestimated, while the northern center of action is shifted towards the east.

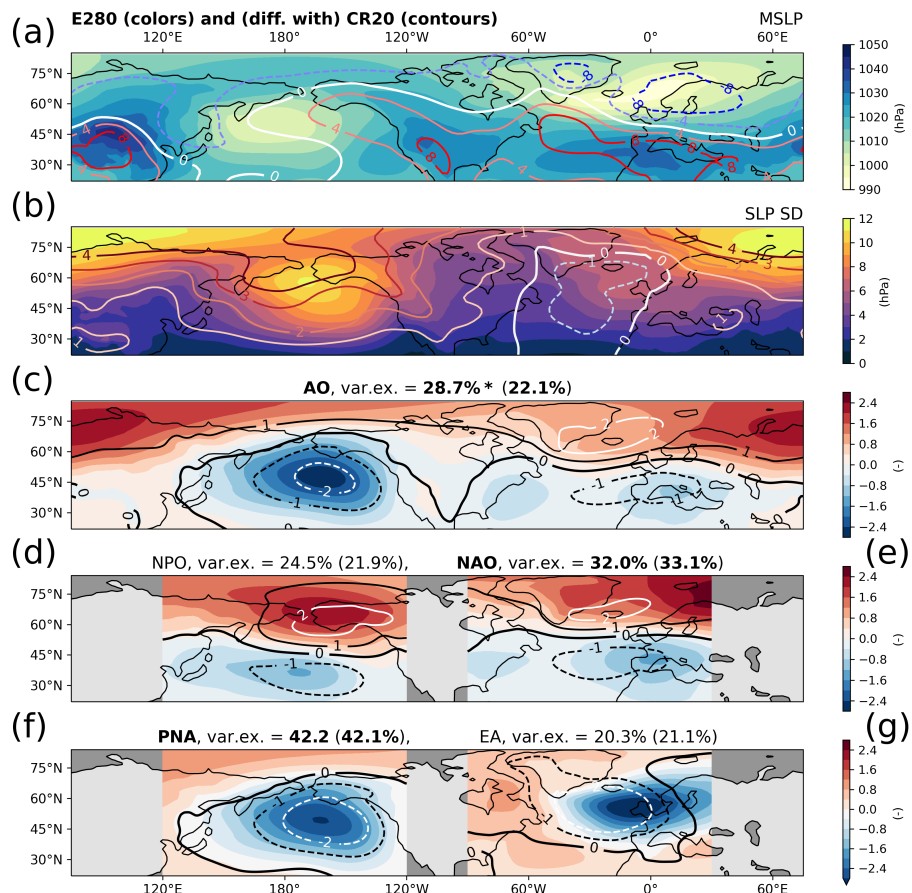

**Figure 1.** (a) January mean SLP and (b) SLP standard deviation (SD) for pre-industrial reference ($E^{280}$; colors) and difference with CR20 reanalysis ($E^{280}$ - CR20; contours). (c-g) January SLP EOFs; (c) Arctic Oscillation (AO), (d) North Pacific Oscillation (NPO), (e) North Atlantic Oscillation (NAO), (f) Pacific-North American (PNA) and (g) East Atlantic (EA) patterns, with $E^{280}$ in colors and CR20 in contours. Percentage variance explained by each $E^{280}$ EOF indicated (CR20 in brackets), * indicates a statistically significant difference between these. In **bold** the (variance of the) leading mode in North Pacific and Atlantic regions.

Again, this can be expected from the differences in total SLP variance as shown in Figure 1b. The second leading mode in the NA is the EA (Figure 1g), explaining 20% (21%) of SLP variance in $E^{280}$ (CR20). The weak but distinct eastern node with opposite sign in the CR20 disappears in the $E^{280}$, likely because the variance shifts more eastward in $E^{280}$. Furthermore, where the CR20 shows that the sign of variability over Greenland is the same sign as in the center of the monopole, the $E^{280}$ shows a clear opposite sign over Greenland. Again, also for this mode the total SLP variance in this region is underestimated. The percentages of variance explained are not statistically different.

In summary, we can say that the $E^{280}$ pre-industrial simulation reproduces the January mean SLP as well as SLP variability from the CR20 reanalysis quite well. The modes in the North Pacific are captured well in spatial extent, amplitude as well

as percentage of variance explained. The modes in the North Atlantic are well reproduced, albeit less accurate, especially regarding amplitudes. However, this is a likely cause of the fact that total SLP variance in the North Atlantic is reduced, which is a known bias in both CMIP5 and CMIP6 models (Eyring et al., 2021).

## 3 Results

### 3.1 Mean winter climate

#### 3.1.1 Mean sea-level pressure and the subtropical jet stream

Figure 2 shows the January MSLP (colors) as well as mean zonal wind at 200hPa (U200, contours). Figure 2a shows the $E^{280}$ MSLP results (similar to Figure 1a but extended southward to 20°S). The January MSLP difference with $E^{560}$ and $Eoi^{280}$ is shown in Figure 2 b and c, respectively. While the MSLP response to $CO_2$ doubling is overall relatively small, the response to mid-Pliocene BCs is substantially larger, especially over the North Pacific. The large increase in MSLP over the NP (+16 hPa) results in a reduced MSLP difference between the NP subtropical high and subpolar low pressure areas, as well as a slight poleward shift of the latitude with the largest MSLP gradient. MSLP, furthermore, decreases over the Arctic in the $Eoi^{280}$ (up to -7 hPa). When comparing with the mid-Pliocene reference simulation $Eoi^{400}$ (annual mean in Baatsen et al. (2022), January mean in Supplement Figure S2), we see that the MSLP response is largely caused by the mid-Pliocene boundary conditions, and not by the $CO_2$ increase. Only the MSLP decrease around Greenland and especially over the Baffin Bay and the Labrador Sea is a clear combination of the response to mid-Pliocene BCs and to elevated $CO_2$, as it can be observed both in the $E^{560}$ and $Eoi^{280}$ results.

Figure 2a also shows $E^{280}$ zonal wind on the 200 hPa isobar. The subtropical jet stream can clearly be identified as a streak of high zonal wind roughly between 25°N and 45°N. The jet is strong and concentrated over East Asia and the North Pacific Ocean. Over the North Atlantic Ocean, two separated jet streams can be seen, roughly between 20-25°N and 45-50°N. The migration of the subtropical jet towards higher latitudes over the NA, compared to the NP, corresponds to the subtropical high and subpolar low pressure being more extended northwards. The $E^{560}$ response in zonal wind is very minimal and limits itself to a few meters per second at most (Figure 2b). Over the NP, a slight increase in wind speeds is found along the center of the jet. Over the Euro-Atlantic sector, the response is relatively weak. The largest increase in zonal wind in the Pacific is 4.4 m/s, which is less than the mean time variation of the $E^{280}$ zonal wind (defined as the SD in time, 5.9 m/s). Generally, the jet stream response to the mid-Pliocene conditions (Figure 2c) is more substantial compared to the response to $CO_2$ doubling, with the largest response over the NP. The subtropical jet stream weakens substantially over the exit of the East Asian jet and continues to weaken over the whole of the NP, with reductions of up to 25 m/s along the jet core, corresponding to 50% over the central NP. Apart from the reduction in jet strength along 30°N, we also see an increase in zonal wind over the northern NP, which indicates that apart from weakening, the subtropical jet shifts or expands polewards. The weakening and widening of the North Pacific subtropical jet is consistent with the slight reduction in SLP difference between the subtropical high and subpolar low. Our findings agree with other PlioMIP2 studies; a strong increase in NP MSLP and simultaneous weakening of the NP jet

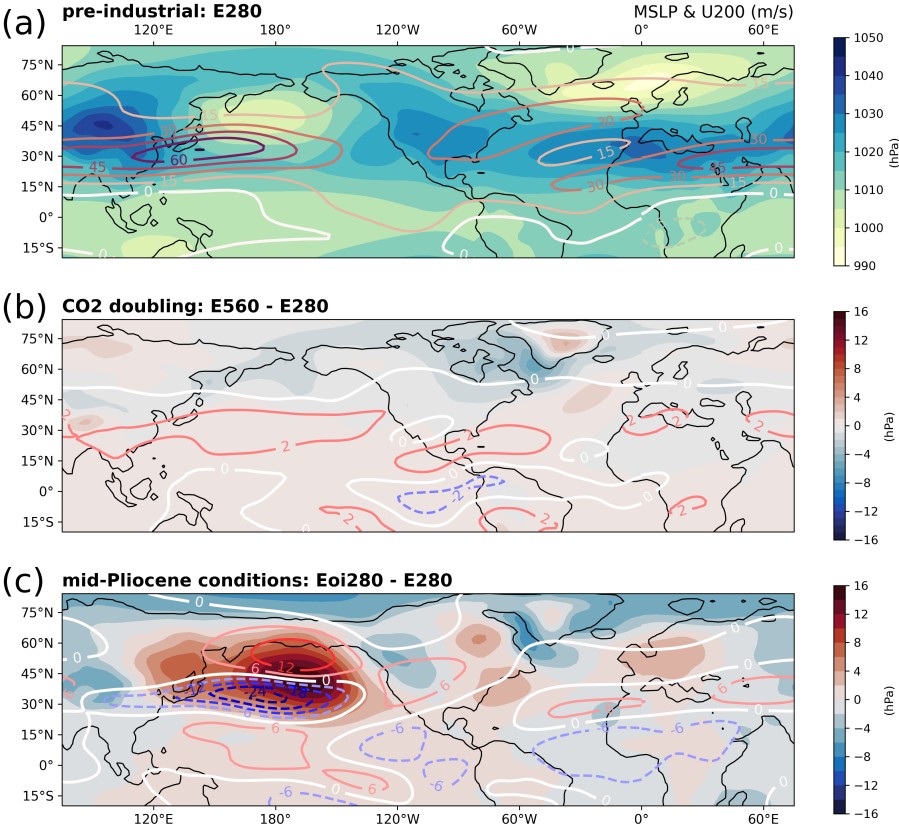

**Figure 2.** January mean sea-level pressure (MSLP, colors) and mean zonal wind at 200 hPa (U200, contours) for (a) pre-industrial reference $E^{280}$, (b) difference with $CO_2$ doubling $E^{560}$ minus $E^{280}$, and (c) difference with mid-Pliocene conditions $Eoi^{280}$ minus $E^{280}$. Note that the increments of the colorbar as well as the labels on the contour lines are slightly different between (b) and (c).

stream in winter is shown in the mid-Pliocene $Eoi^{400}$ simulations with CCSM4-UoT (Menemenlis et al., 2021), and a weaker NP jet stream in the annual mean is reported in $Eoi^{400}$ simulations with HadCM3 and CESM2 (Hunter et al., 2019; Feng et al., 2022).

### 3.1.2 Surface temperatures and precipitation

Figure 3 shows the January mean surface air temperature (SAT, colors) as well as precipitation (contours). Figure 3a presents January SAT for the pre-industrial reference $E^{280}$. Arctic cold spreads far south over the continental regions, specifically Siberia and northern Canada. SAT distributions over land and the ocean are largely as expected from present-day and historical observations. The SAT difference ($E^{560}$ - $E^{280}$, Figure 3b) show that most continental areas warm slightly more compared to oceans at similar latitudes. Furthermore, a clear Arctic amplification signal can be observed. This response is expected with increasing atmospheric $CO_2$ levels (Serreze and Barry, 2011). Temperatures increase more than 15°C over the Iceland,

Norwegian and Barents Seas. Extensive warming is furthermore found in the Labrador Sea, Bering Strait and Okhotsk Sea. The SAT response for the Eoi[280] is in many ways similar to the SAT response for the $CO_2$ doubling (Figure 3c). An Arctic amplification response is present even without any increase in $CO_2$ levels. Again, we see a large warming over the Greenland Sea and Okhotsk Sea, as well as over the Baffin Bay and Labrador Sea. In the Eoi[280] results, we see a warmer region over the western and central North Pacific and a large area of cooling over western North America and the land bridge that is now the Bering Strait. This SAT dipole is consistent with the MSLP increase over the North Pacific and decrease over western North America as seen in Figure 2c.

Figure 3a also shows the January mean precipitation for the E[280]. The midlatitude storm tracks over the NP and NA basins are clearly visible. A band of heavy precipitation exists in the tropics, which represents the Intertropical Convergence Zone (ITCZ). A double ITCZ is present over the Pacific Ocean, which is a known model bias and reported before in Baatsen et al. (2022). The precipitation response to $CO_2$ doubling (Figure 3b) is minimal; precipitation is slightly intensified over the tropics, and a concentrated precipitation increase is seen South of Greenland, which is mainly caused by a retreat of sea-ice in that area (not shown), agreeing with the local SAT increase and MSLP decrease. The precipitation response to mid-Pliocene boundary conditions (Figure 3c) is more substantial compared to the precipitation response to $CO_2$ doubling (similar to the MSLP and U200 response). The Pacific ITCZ shifts northwards, although the southern branch weakens. The northward ITCZ shift is consistent with most PlioMIP2 Eoi[400] simulations (Han et al., 2021), and has been connected to a weakening of the amplitude of ENSO variability (Pontes et al., 2022). Furthermore, there is a substantial drying over the west-equatorial Pacific (WEP), accompanied by a large increase in precipitation over the Indian Ocean. We associate this with a westward expansion of the Walker circulation and an intensification of the Indian monsoon, which was reported before in CCSM4-Utr in Baatsen et al. (2022), and seen in most other PlioMIP2 Eoi[400] simulations (Han et al., 2021). Precipitation also intensifies over the equatorial Atlantic, which might be related to the intensification of the West African monsoon as shown in most PlioMIP2 Eoi[400] simulations (Berntell et al., 2021). Furthermore, we observe a drying in the NP subtropics, which mainly results in reduced storm track activity over the northeast Pacific Ocean. Lastly, precipitation increases south of Greenland and in Labrador sea, which is - similar to the $CO_2$ doubling response - related to sea-ice retreat (not shown).

## 3.2 Sea-level pressure variability

Following the large changes in mean winter climate in the simulation under mid-Pliocene conditions, we will now analyse the changes in atmospheric variability by means of EOF analysis of the SLP in, respectively, the NH, NP and NA sectors.

### 3.2.1 Variability in response to $CO_2$ doubling

Figure 4a shows the E[560] SLP SD and SLP SD difference with E[280] in contours. In most of the NH, there are no notable changes in SLP SD. Only in the center-east North Pacific, there is an increase in SLP SD in the $CO_2$ doubling simulation with regards to the pre-industrial (up to 30%).

The spatial pattern as well as regional amplitude of all of the modes of variability are very similar in the E[560] and E[280] (Figures 4b-f). As in the pre-industrial, the leading mode in the Pacific (Atlantic) is the PNA (NAO) mode. In the AO, the

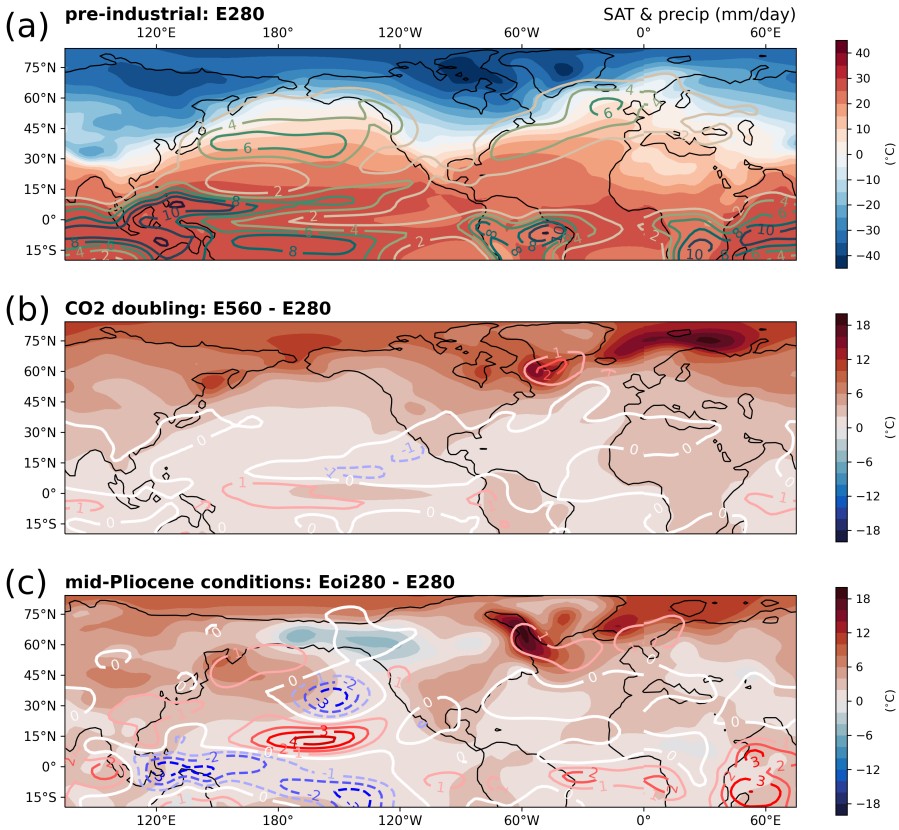

**Figure 3.** January mean mean surface air temperature (SAT, colors) and mean precipitation (precip, contours), for (a) pre-industrial reference $E^{280}$, (b) difference with $CO_2$ doubling $E^{560}$ minus $E^{280}$, and (c) difference with mid-Pliocene conditions $Eoi^{280}$ minus $E^{280}$.

amplitude of the North Pacific center of action is slightly increased, while the amplitude of the North Atlantic-Siberian center of action is reduced. The AO explains more of the total SLP variance in the $CO_2$ doubling (38.9%) compared to the pre-industrial (28.7%), which is a statistically significant difference. Both results are consistent with the fact that there is simply more SLP variance over the North Pacific (variance defined as square of the SLP SD, Figure 4a).

The two leading modes in the NP explain more SLP variance in the $CO_2$ doubling simulation compared to the pre-industrial (75.2% over 66.7%, Figures 4c and e). This is mainly because the leading PNA mode becomes more dominant, explaining 57.2% of the total SLP variance in this region, which is a lot more compared to the $E^{280}$ (42.2%, significant difference). The second leading mode, the NPO, explains a bit less of the total SLP variance (also significant difference). These results are consistent with Chen et al. (2018), that show a strengthening of PNA intensity under a strong future warming scenario in an ensemble of 35 CMIP5 models. The two leading modes in the NA are shown in Figures 4d and f and the percentage of variance explained remains the same in the $E^{280}$ and $E^{560}$ (no significant difference). The dipole shape of the NAO is shifted slightly polewards, and the southern node is shifted slightly westward. For the EA, the separation between positive and negative

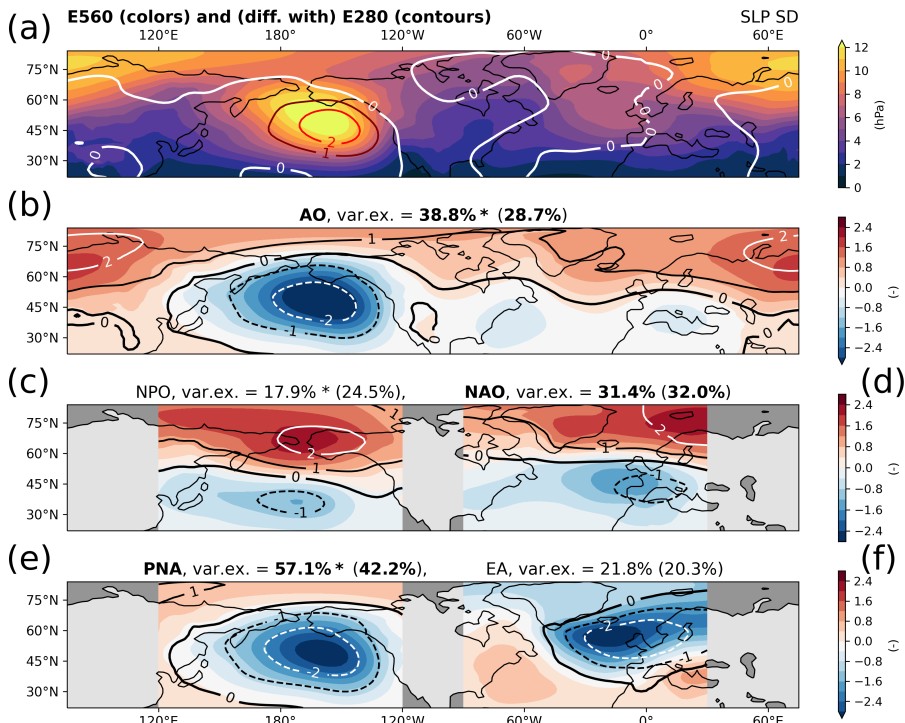

**Figure 4.** (a) January SLP standard deviation (SD) in $CO_2$ doubling (E$^{560}$; colors) and difference with pre-industrial (E$^{560}$ - E$^{280}$; contours). (b-e) January SLP EOFs; (b) Arctic Oscillation (AO), (c) North Pacific Oscillation (NPO), (d) North Atlantic Oscillation (NAO), (e) Pacific-North American (PNA) and (f) East Atlantic (EA) patterns. E$^{560}$ in colors and E$^{280}$ reference in contours. In **bold** the leading mode in each sector. Percentage variance explained by each E$^{560}$ EOF indicated (E$^{280}$ in brackets), * indicates a statistically significant difference between these.

amplitude (or red and blue) moves over the Canadian archipelago in the E$^{560}$, while it moves eastward and poleward from the southern tip of Greenland in the E$^{280}$.

### 3.2.2 Variability in response to mid-Pliocene boundary conditions

Figure 5a shows the Eoi$^{280}$ SLP SD and SLP SD difference with E$^{280}$ in contours. In general, there is a decrease in SLP SD in the mid-Pliocene simulation. However, there is a substantial decrease in SLP SD over the northeastern North Pacific (up to -40%), while there is a small increase in SLP SD over the Canadian Arctic. Furthermore, there is a decrease in SLP SD over the Scandinavian Arctic and eastern Siberia. No substantial changes are observed over the Atlantic. The changes in SLP SD in the Eoi$^{280}$ are very similar to the SLP SD changes in the Eoi$^{400}$ (see Supplementary Material Figure S2).

The spatial patterns of the modes of SLP variability are similar for the Eoi$^{280}$ and E$^{280}$ (Figures 5b-f). Still, some distinct differences can be seen. The leading mode in the NA is still the NAO, but the leading mode in the NP becomes the NPO, whereas it is the PNA in the E$^{280}$ and E$^{560}$. The spatial pattern of the AO over the North Pacific changes quite drastically,

describing more of a dipole rather than a large single center of action. The AO mode thus becomes more zonal or annular. The percentage of SLP variance explained is similar in the Eoi$^{280}$ (28.3%) compared to the E$^{280}$ (28.7%, no significant difference).

The two leading modes in the NP together explain slightly less of the total SLP variance in comparison to the pre-industrial (63.9% over 66.7%, Figures 5c and e). The leading mode becomes the NPO, explaining almost double the variance that it explains in the E$^{280}$ (45.5% over 24.5%, significant difference). The spatial pattern is very similar, but differences are that the whole dipole is shifted slightly polewards, and that both centers of action are more spread out spatially, thus representing SLP variations over a larger area. The second leading mode in the NP becomes the PNA, explaining only 18.4% of the SLP variance compared to 42.2% in the pre-industrial (significant difference). The spatial extent of the monopole is slightly shifted polewards, and the region with largest amplitude extents more over the northwestern North American continent. The dominance of the NPO over the PNA is consistent with the change in SLP variance (or squared SLP SD) over this region (Figure 5a).

The leading mode in the NA is the NAO (Figures 5d) and explains almost the same amount of SLP variance (32.7% over 32.2%, no significant difference). The centers of action of the dipole are shifted slightly polewards. Especially the southern node is more centred over northwestern European mainland, while the northern node is retreated polewards and weakened. The EA pattern (Figures 5f) explains slightly less variance in the Eoi$^{280}$ (17.5% over 20.3%, no significant difference). The center of action is shifted polewards and spread out further towards the northeast. The southern separation between negative and positive amplitude is shifted northwards over the European mainland. Over the western part of the sector, the mode resembles a dipole with a strong northern node and weak southern node.

### 3.2.3 Correlations between modes

Figure 6a presents the PC correlation matrix for the E$^{280}$ as well as the CR20 reanalysis. The correlations are very similar in both cases, as can be observed from the high level of symmetry with respect to the diagonal. Both show a strong correlation between the AO and NAO, and between the AO and PNA. However, in the CR20 reanalysis these two correlations are of similar magnitude, while the E$^{280}$ shows a much stronger correlation of the AO with the PNA than with the NAO. This is consistent with the lower (higher) SLP variance over the North Atlantic (North Pacific) in the E$^{280}$ compared to the CR20 as shown in Figure 1b. The strongest correlation between basins is between the PNA and NAO, which can be expected since both are strongly correlated with the AO. The NPO does not correlate strongly with any other mode in both the E$^{280}$ (only a weak correlation with the AO) and CR20.

The correlation matrix for the E$^{560}$ shown in Figure 6b shows great similarities with the E$^{280}$. However, a notable difference is the even stronger correlation between the AO and PNA, of 0.98, that is consistent with the increase in SLP variance explained by the PNA in the E$^{560}$ (see Figure 4e). Interestingly, the AO is more strongly (anti)correlated with the EA than the NAO, which is not very obvious based on the similarities of the spatial patterns only. The AO shows the strongest correlation with the EA (in the NA) and PNA (in the NP), and simultaneously the EA and PNA show a stronger correlation. Again, the NPO does not strongly correlate with another mode.

The Eoi$^{280}$ correlation matrix shown in Figure 6c shows some notable differences with respect to the E$^{280}$ and E$^{560}$. The correlation between the AO and PNA becomes statistically insignificant. The correlation between the AO and NPO is 0.96 in

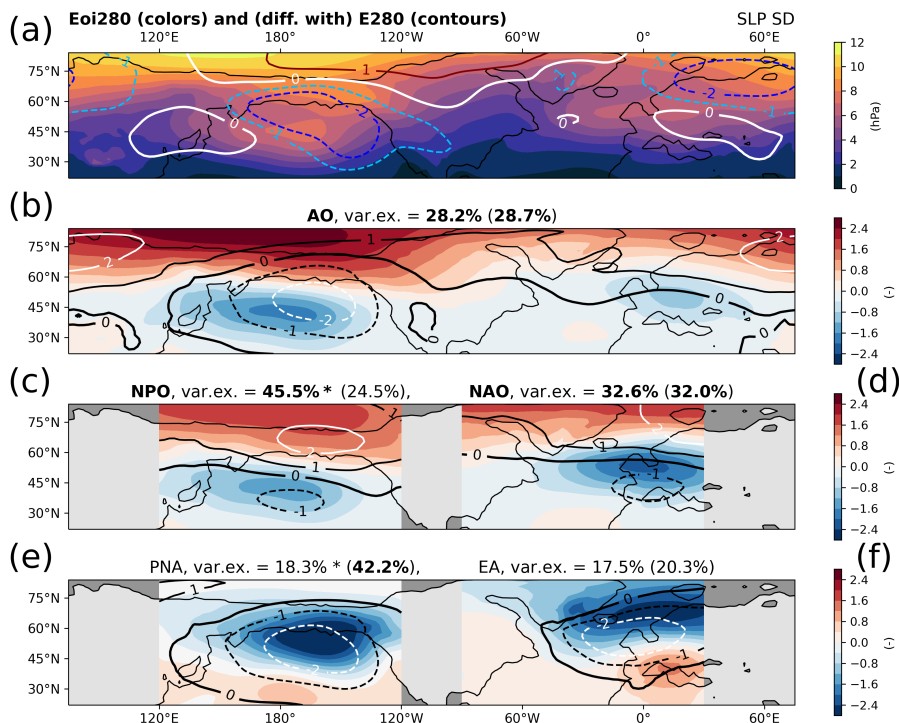

**Figure 5.** a) January SLP standard deviation (SD) in mid-Pliocene conditions (Eoi[280]; colors) and difference with pre-industrial (Eoi[280] - E[280]; contours). (b-e) January SLP EOFs; (b) Arctic Oscillation (AO), (c) North Pacific Oscillation (NPO), (d) North Atlantic Oscillation (NAO), (e) Pacific-North American (PNA) and (f) East Atlantic (EA) patterns. Eoi[280] in colors and E[280] reference in contours. In **bold** the leading mode in each sector. Percentage variance explained by each Eoi[280] EOF indicated (E[280] in brackets), * indicates a statistically significant difference between these.

the Eoi[280], while being very small or insignificant in the other simulations. The AO shows the strongest correlation with the NAO (in the NA) and NPO (in the NP), agreeing with the earlier result that it represents a more annular mode. There is not a very strong correlation between the NPO and NAO, however, as well as between the EA and PNA. In the Eoi[280], the PNA becomes the mode that does not correlate strongly with any other mode (except a small but significant anti-correlation with the EA).

**3.3   North Pacific variability and the jet stream in response to mid-Pliocene boundary conditions**

In this section, we focus on the Eoi[280] and E[280] results and investigate the impact of the changes in SLP variability on the jet stream in the North Pacific.

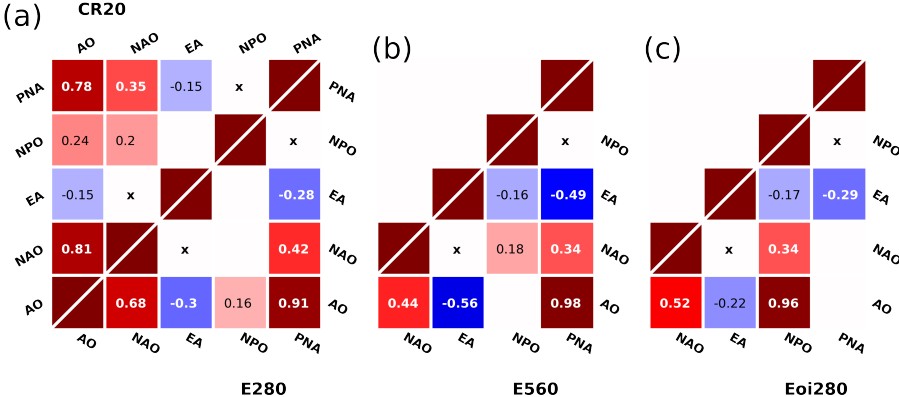

**Figure 6.** Correlation coefficients between Principal Components (PCs) time series belonging to the AO, NAO, EA, NPO and PNA modes. For CR20 (a, top left), $E^{280}$ (a, bottom right), $E^{560}$ (b) and $Eoi^{280}$ (c). **Bold** font if the correlation coefficient is statistically significant at p<0.005, normal font when p<0.05, no correlation shown (white) when p>0.05. **x** for orthogonal constructed modes (correlation 0 by definition).

### 3.3.1 SLP variability and jet variations in the central North Pacific

The temporal behaviour of the $E^{280}$ and $Eoi^{280}$ central North Pacific jet stream is shown in Figure 7a and b, by means of a
Hovmöller diagram of the zonal mean zonal wind at 200 hPa averaged over 160°E - 220°E. The $E^{280}$ shows a strong and focused jet with little latitudinal variations. On the other hand, the $Eoi^{280}$ shows a generally weaker jet with a great latitudinal variation.

The jet intensity versus the PNA index (defined as the PC of the EOF), as well as jet latitude versus NPO index are shown in Figure 7c and d, respectively. Figure 7c shows that the $E^{280}$ jet intensity correlates strongly (0.92) and significantly with the PNA index. Interestingly, the $Eoi^{280}$ jet shows the same strong (0.92) and significant correlation, but with a distribution of much lower wind speeds, and much smaller $R^2$. This means that in both simulations, the jet intensity is linked to the phase of the PNA.

Figure 7d shows the relation between the jet latitude and the NPO index. In both simulations, a clear anti-correlation between the jet latitude and the NPO index is apparent. In the $E^{280}$, the jet is located between 30-40°N. In the $Eoi^{280}$, the jet latitude covers a much larger range, and the histogram reveals a less unimodal distribution, with two weak peaks at 30-40°N and 45-55°N. The scatter plot shows that in negative NPO phases, in most years the max zonal wind is found at higher latitudes, but in some Januaries the jet latitude is found between 20-30°N. It suggests the existence of a state with two jets, where generally the northbound jet is stronger. The histogram of NPO index values furthermore shows a negative skew, coinciding with this split jet or more northward jet in the $Eoi^{280}$. The $E^{280}$ distribution is slightly skewed towards positive NPO values, which implies a focused jet in the subtropics. The fact that a split jet exists in the NPO- phase implies that the distribution of jet latitudes is not linear, which suggests that a linear fit might not be the best metric to capture the correlation.

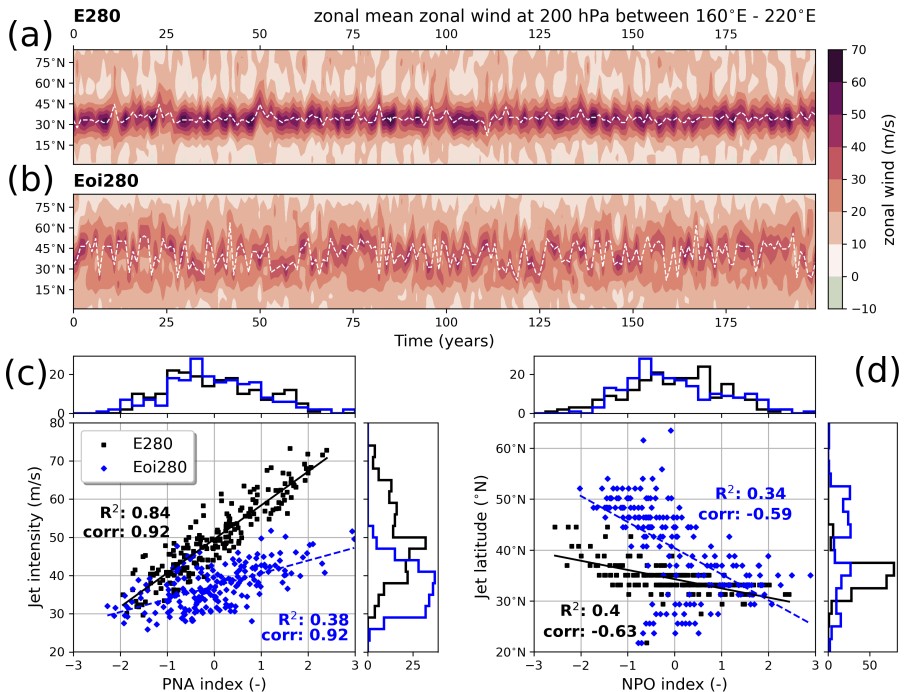

**Figure 7.** (a,b) Hovmöller diagrams showing zonal mean zonal wind at 200 hPa in the $160°E$ - $220°E$ mean for every January for the $E^{280}$ (a) and $Eoi^{280}$ (b). The white dashed line follows the maximum of zonal wind. (c) Scatter plot including histograms of counts for the PNA index (i.e. principal component) versus the jet intensity (defined as max zonal wind). $R^2$ of linear fit and correlation coefficient are shown. For $E^{280}$ (black squares) and $Eoi^{280}$ (blue diamonds). (d) Same, but for NPO index versus jet latitude (defined at latitude of max zonal wind).

The temporal behavior of the jet as well as the correlation between the PNA and NPO, and the jet intensity and latitude suggest the following. In the $E^{280}$, the jet is mainly controlled by the PNA, with a high variation in jet intensity but not in latitude, while in the $Eoi^{280}$, the jet is mainly controlled by the NPO, with a high variation in the jet latitude and a tendency towards a strong northwards and weak southward jet, with less variation in jet intensity. Our findings agree with literature, where the PNA is connected to jet enhancement and displacement in the east-west direction, while the NPO is more connected to the displacement of the jet in the north-south direction (see e.g. Linkin and Nigam, 2008). The dominance of the PNA (NPO) in the $E^{280}$ ($Eoi^{280}$) corroborates our previous findings on the percentage of SLP variance explained by both modes (see Figure 5). Analyses using the global zonal mean zonal wind and the AO mode (Figure S3), as well as with the PNA and jet latitude, and NPO with jet intensity (Figure S4), are included in Supplementary material.

### 3.3.2 The phases of the mid-Pliocene NPO

The state of the NP jet stream in the NPO+ and NPO- phases in the $E^{280}$ and $Eoi^{280}$ is shown in Figure 8, where the NPO+ and NPO- phases are defined as the ten average Januaries corresponding to the ten highest and ten lowest NPO index values,

respectively. Figure 8 shows the zonal wind and 2PVU contour at 200 hPa on the right and the zonal mean between 160°E - 220°E on the left for both phases and the two simulations. The E$^{280}$ NPO- phase in Figure 8a shows a strong jet around 30°N that deflects southward, with a sharp 2PVU gradient along the jet core in the vertical in the central North Pacific. The E$^{280}$ NPO+ phase shown in Figure 8b shows a slightly weaker jet between 35 - 45°N, agreeing with a 2PVU gradient that is less sharp, that moves more northward over the North Pacific.

The Eoi$^{280}$ NPO- phase is shown in Figure 8c and shows similar zonal wind patterns and 2PVU contours as in the E$^{280}$, but at lower zonal winds (max 43.0 m/s in the central North Pacific zonal mean over 48.2 m/s in the E$^{280}$). The NPO+ phase in the Eoi$^{280}$ (Figure 8d) is different from the E$^{280}$, clearly showing the existence of two jets over the NP, both on the 200 hPa isobar as well as in the vertical cross section. The northern jet (40.2 m/s, between 45-55°N) is stronger than the southern jet (27.2 m/s, between 15-25°N), confirming the suggestion of the existence of two jets based on the results in Figure 7d. The southern jet is located at lower pressures (or higher in the atmosphere) than the northern jet, although both jet cores follow the sharpest gradients of the 2PVU curve. The existence of this split jet over the NP indicates a higher level of anticyclonic wave breaking on the northward side of the mean subtropical jet in comparison to the E$^{280}$, effectively deflecting the jet towards higher latitudes. The relationship between the NPO and Rossby wave breaking is described in more detail in Rivière (2010). The fact that in all cases the strongest jets are present where the meridional PV gradient is the strongest (see Figure 8 left panels) may indicate that the atmosphere is approximately in thermal wind balance in the January monthly mean. The nearly constant 2PVU height over latitude between the two jets in the Eoi$^{280}$ NPO+ (Figure 8d) furthermore indicates a high level of meridional mixing in the atmosphere between these two jets. The SLP and SAT anomalies associated with the NPO- and NPO+ phases are shown in Supplementary Material Figure S5.

### 3.3.3 Tropical Pacific convection as an explanation

In this section, we present a hypothesis that could explain the substantial reduction in NP SLP variability under mid-Pliocene boundary conditions (Eoi$^{280}$), and subsequent relative dominance of the NPO and more pronounced north-south displacement of the NP jet stream. The dynamical explanation is as follows: in the Pacific sector, a weakening of the tropical convection signal (related to shifts in the mean state and due to reduced ENSO variability) leads to reduced upper-tropospheric Rossby wave activity in the extratropics, which ultimately leads to a jet stream that is weaker and less variable in east-west extent, which in turn explains the reduced variance of SLP at the surface. We highlight the steps of this mechanism in more detail below.

1a. Shifts in the tropical mean state reduce mean precipitation in the west-equatorial Pacific (WEP), specifically the westward shift of the upward branch of the Walker circulation, as well as the northward shift of the ITCZ (Figure 3c).

1b. Reduced ENSO variability reduces WEP precipitation variability (Figure 9a, explained in more detail below). Taking precipitation as a proxy for latent heating or convection in the tropics, both 1a and 1b lead to a reduced convection signal in the WEP.

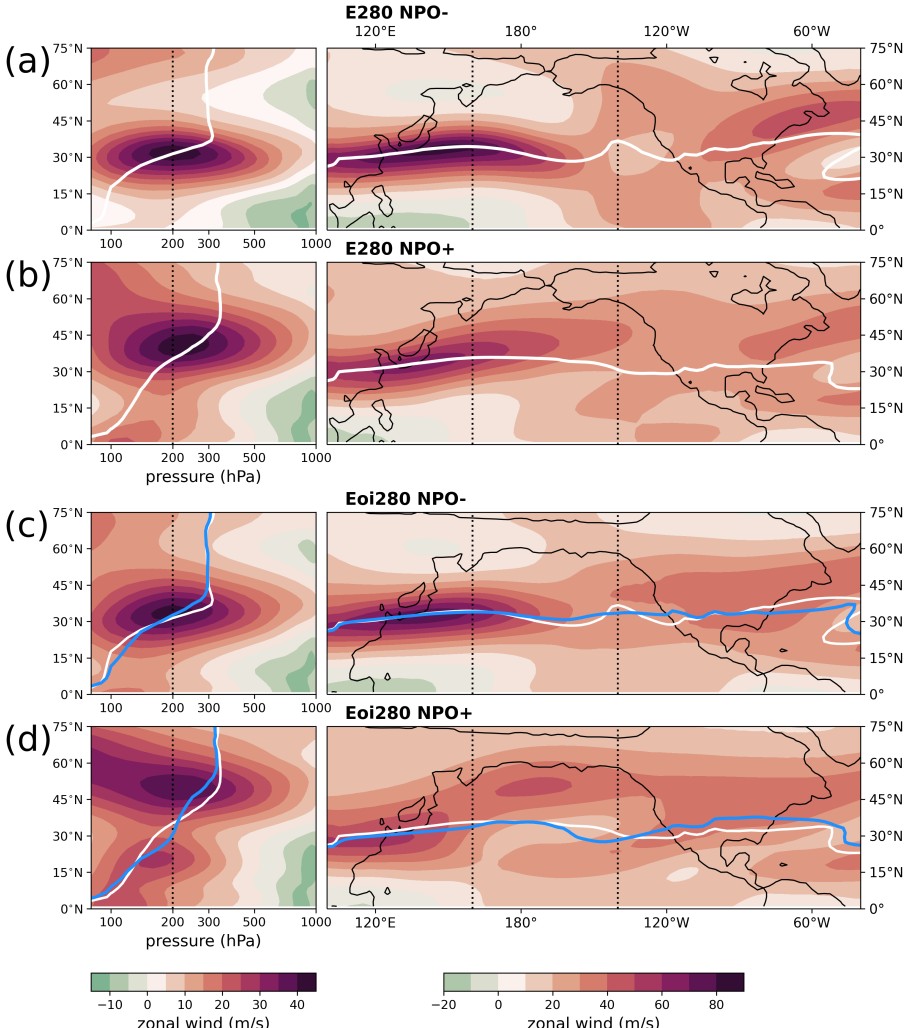

**Figure 8.** Zonal mean zonal wind between $160°$E - $220°$E (left) and zonal wind at 200 hPa (right). For the $E^{280}$ (a, b) and $Eoi^{280}$ (c, d), representing the NPO- phases (a, c) and NPO+ phases (b, d). Contour line indicates the 2PVU contour, in white for the $E^{280}$ (repeated in c, d) and blue for $Eoi^{280}$. Note that the colorbar range is slightly different for the left and right panels.

2. Reduced convection in the WEP substantially weakens Rossby wave activity in the upper troposphere in the northern extratropics, primarily over the East Asia jet exit and western North Pacific (more detail below).

3. Since the jet stream acts a wave guide for Rossby waves (e.g. Branstator, 2002), reduced upper-tropospheric Rossby wave activity over East Asia leads to a North Pacific jet stream that is less variable in strength (Figure 7c) and more variable in latitudinal location (Figure 7d)

4a. A jet stream weak in strength correlates with the negative phase of the PNA (Figure 7c), which implies a positive SLP anomaly over the North Pacific, and agrees with the MSLP difference in Figure 2c.

4b. A jet less variable in strength implies a relatively weaker PNA, while a jet more variable in latitude implies a relatively stronger NPO (according to the correlations in Figure 7c and d), which agrees with our findings in terms of SLP variability in Figure 5c and e.

Figure 9a shows a scatter plot of the precipitation in the WEP as a function of the Nino3.4 index. In all of our simulations, the correlation between these two quantities is strong (>0.72), positive and statistically significant. We found that the precipitation in the WEP showed the strongest regression of the Nino3.4 indexc with global precipitation (see Supplementary material Figure S6). The mechanism linking ENSO variability to convection in the WEP is established before (see e.g. Hoskins and Karoly, 1981), so it is not surprising that the regression (or: linear slope) between the two is similar for all simulations. Apart from the mean precipitation signal being lower in the Eoi$^{280}$ (1a.), the precipitation variability is also reduced (1b.), which is caused by the reduction in ENSO variability. The SD of the Nino3.4 index (also called the ENSO amplitude) reduces substantially in response to the mid-Pliocene boundary conditions (0.4°C in the Eoi$^{280}$; 1.1°C in the Eoi$^{280}$), but does not change in response to $CO_2$ doubling (1.1°C in the E$^{560}$). A similar reduction is observed in the Eoi$^{400}$ simulation (0.4°C).

The influence of ENSO variability on North Pacific atmospheric variability (specifically the PNA) has been described before (see e.g. Mo and Livezey, 1986; Yeh et al., 2018; Domeisen et al., 2019). In short, the mechanism in the present-day climate is as follows: tropical Pacific SST anomalies lead to anomalous convection, and with that force anomalies in the Hadley circulation (Hoskins and Karoly, 1981); this anomalous vertical motion leads to upper-tropospheric divergence in the tropics, and convergence in the subtropics, which in turn results in vorticity anomalies (Mo and Livezey, 1986); this upper-level vorticity is a source of Rossby wave activity over the North Pacific (Mo and Livezey, 1986; Nie et al., 2019). In summary, Rossby waves connect the tropical heating signal to the atmospheric variability in the higher latitudes, and thus connect ENSO variability to NP SLP variability. This connection is shown in Figure 9b, which shows the regression (linear slope) between the WEP precipitation and the SLP (colors) and U200 (contours) anomalies (summarizing 2. - 4b.). The results show that the E$^{280}$ simulation is able to capture the connection between tropical convection and North Pacific SLP and jet stream variations. This mechanism breaks down in the Eoi$^{280}$ (Figure 9c). The regression pattern of WEP precipitation with SLP shows more of a dipole (reminiscent of the NPO pattern), and the regression pattern of WEP precipitation with the jet variability moves such that it represents more latitudinal variation in the jet stream. Note that while the regression has similar, or stronger, values in the Eoi$^{280}$ compared to E$^{280}$, the WEP precipitation values are substantially lower, thus leader to lower values for SLP and U200 variations as well. More extensive analysis on the connection between tropical convection and Rossby wave activity in our simulations can be found in the Supplementary material section 5, Figures S7 - S9.

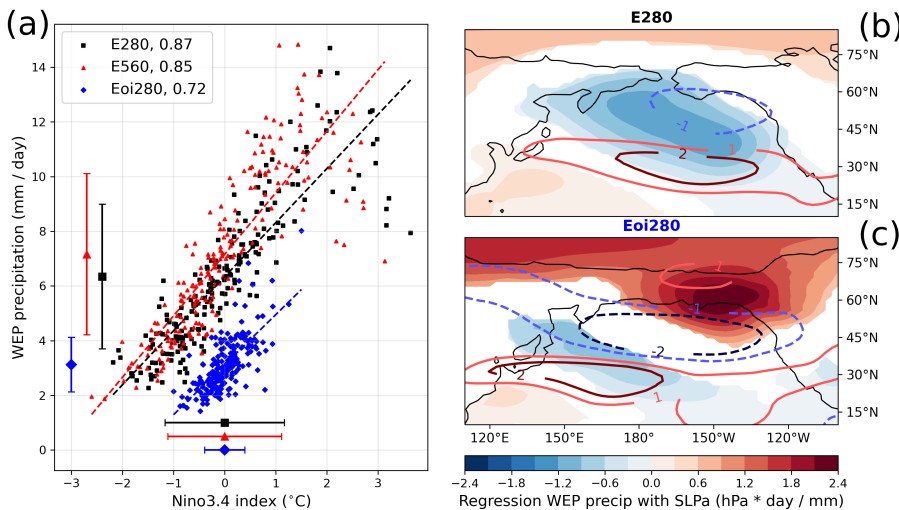

**Figure 9.** (a) Scatter plot of Nino3.4 index versus average precipitation in the West equatorial Pacific (WEP precip) for $E^{280}$ (black squares), $E^{560}$ (red triangles) and $Eoi^{280}$ (blue diamonds). Including linear regression, correlation coefficient in the caption, and mean and standard deviation values indicated range along the axes. (b, c) Regression (linear slope) between WEP precipitation and SLP anomalies (colors), and WEP precipitation and zonal wind at 200 hPa (contours), for $E^{280}$ (b) and $Eoi^{280}$ (c). Regression only shown when correlation is significant ($p<0.05$).

## 4  Discussion

### 4.1  Physical and dynamical interpretation

#### 4.1.1  Changes in variability and changes in the winter mean state

Under $CO_2$ doubling, our simulations show a strengthened PNA, consistent with a general increase in SLP SD over the North Pacific. This is consistent with a study by Chen et al. (2018), that show a strengthening of PNA intensity under a strong future warming scenario in an ensemble of 35 CMIP5 models. They link it to a strengthening of tropical Pacific precipitation response, caused by enhanced central equatorial Pacific SST variability. Our $E^{560}$ simulation does show a slight enhancement of tropical Pacific precipitation (see Figure 3b and 9a). Furthermore, we find a slight intensification of the NP subtropical jet stream (Figure 2b), which according to the relationship described in Figure 7c is related to an enhanced PNA.

As a response to mid-Pliocene boundary conditions, we find a relative strengthening of the NPO and simultaneous weakening of the PNA, with a general decrease of SLP SD over the North Pacific. The NPO becomes dominant even on a hemispheric scale where it correlates strongly and significantly with the AO. The North Pacific jet becomes weaker (consistent with other PlioMIP2 studies; Hunter et al. (2019), Menemenlis et al. (2021) and Feng et al. (2022)) and more variable in latitude, which is correlated to NPO variability. The North Pacific winter climate is oscillating between states either with a relatively strong jet (NPO-), or with a split jet that has a strong northward jet and weak southward jet (NPO+). In the winter mean, there is a

strong increase in MSLP over the North Pacific, weakening the Aleutian low (consistent with Menemenlis et al., 2021). In the previous section, we have described a hypothesis by which changes in the tropical Pacific could explain the changes in North Pacific atmospheric variability due to mid-Pliocene boundary conditions.

To summarize: in both simulations, the changes in NH atmospheric variability are mainly related to changes in the tropical Pacific mean state. In response to $CO_2$ doubling, the changes in this mean state are small, as are changes to the NP variability. In response to mid-Pliocene boundary conditions, the changes in the tropical mean state are substantial, leading to large changes in the variability of the jet stream and the SLP patterns. The changes in the NA mean state, as well as in NA variability, are small.

### 4.1.2 The role of mean surface temperature response

Qualitatively, the SAT response to increased $CO_2$ and to the mid-Pliocene BCs is similar. However, Baatsen et al. (2022) show that for the CCSM4-Utr, the majority of the high-latitude Eoi[280] warming is due to a reduced planetary albedo, due to both reduced ice sheet and reduced sea-ice cover, while in the E[560] warming is mostly due to increased greenhouse gas concentrations (consistent with other PlioMIP2 studies; Chandan and Peltier, 2018; Burton et al., 2023). Furthermore, the warming response to $CO_2$ is mostly zonally uniform with distinct Arctic amplification, while the response to mid-Pliocene BCs is more regional, especially over the North Pacific, where SAT increases over the western part and decreases over the eastern part.

Menemenlis et al. (2021) find a similar SAT response in their mid-Pliocene simulations with CCSM4-UoT as our Eoi[280] response in Figure 3c and interpret it as a cool phase of the PDO, which is related to a weakened Aleutian low. While the connection between the PDO and Aleutian low is clear from literature (see e.g. Hurwitz et al., 2012; Simon et al., 2022), assessing the links with (multi)decadal variability is outside of the scope of this work, and Pacific (multi)decadal variability in the PlioMIP2 ensemble will be investigated in future research. We interpret the weakening of the Aleutian low in our Eoi[280] simulation as a weakening of the dominance of the PNA in the pre-industrial, which simultaneously allows the NPO to exert more influence on the North Pacific winter climate. As Linkin and Nigam (2008) explain, the NPO can be considered a Pacific analogue to the NAO in the NA. In the present-day climate, the amplitude of the NAO is large and the NA jet relatively weak, while the NPO amplitude is small and the NP jet relatively strong. In our mid-Pliocene simulations, the NP jet weakens up to 50% ánd the NPO becomes the dominant mode of variability in the NP. This implies that - like the NAO in the present day NA - SLP anomalies related to NPO variability strongly relate to the latitudinal displacement of the NP jet. This is what we see in Figure 7d. The split jet condition over the NP, related to the NPO+ phase in the mid-Pliocene simulation (Figure 8d), is also reminiscent of the dynamic behaviour of the jet in the present-day NA, corroborating the NAO analogue even further.

## 4.2 A climate variability point of view

How do the observed mid-Pliocene changes in North Pacific winter variability link to other modes of variability? Apart from the dominant Pacific teleconnection between canonical ENSO, PDO and PNA, the NPO has been linked to the North Pacific Meridional Mode (NPMM, Chiang and Vimont, 2004; Linkin and Nigam, 2008), as well as to the central Pacific ENSO, also

known as El Niño Modoki or warm pool El Niño (Furtado et al., 2012; Ding et al., 2022). In the last section of the Results, we
show that the North Pacific SLP variability relates to ENSO variability, but not specifically to which ENSO flavour, nor how
other modes of variability could be involved. We know that in the Eoi$^{400}$ simulation with CCSM4-Utr the canonical ENSO
variability reduces (Oldeman et al., 2021), as well as the variability of the PDO (Baatsen et al., 2022, referred to as Pacific
Multidecadal Variability or PMV). The reduced canonical ENSO and PDO variability could be connected to the reduced PNA
variability in the North Pacific winter. Oldeman et al. (2021) also show a slight relative increase in the number of central
Pacific ENSO events over canonical ENSO events in the CCSM4-Utr Eoi$^{400}$, consistent with the relative strengthening of the
NPO. However, the mid-Pliocene NPMM is shown to be substantially weaker in CCSM4-Utr compared to the pre-industrial
by Pontes et al. (2022, extended data). Recently though, Stuecker (2018) showed that the PNA and PDO are also connected
to the NPMM and central Pacific ENSO, especially on low frequencies. Our results consistently show a total reduction in
the Pacific oceanic and atmospheric variability in the CCSM4-Utr mid-Pliocene simulations, with a clear reduction of the
dominant teleconnection (being PNA, PDO and canonical ENSO) and a relative increase of the prominence of the second
leading teleconnection (being NPO, NPMM and central Pacific ENSO).

### 4.3 Sensitivity to the mid-Pliocene boundary conditions and CO$_2$

In this study we have investigated mid-Pliocene NH winter variability, and separated the response to either CO$_2$ or to mid-
Pliocene boundary conditions other than CO$_2$. We find large differences between both climate forcings. In this section we want
to discuss how sensitive the responses found are to the combination of mid-Pliocene boundary conditions and elevated CO$_2$.
Furthermore, we explore which of the specific boundary condition changes in the Eoi$^{280}$ could explain the observed response.

Firstly, we want to address the non-additivity of the responses to changes in boundary conditions, or elevated CO$_2$. To
investigate this, we briefly study the results of an additional Eoi$^{560}$ simulation (described in Baatsen et al. (2022)), which
represents a simulation with mid-Pliocene boundary conditions and a doubled pre-industrial CO$_2$ level. With this simulation,
we can assess the response to mid-Pliocene boundary conditions at elevated CO$_2$ conditions (Eoi$^{560}$ - E$^{560}$), as well as the
response to a CO$_2$ doubling in mid-Pliocene conditions (Eoi$^{560}$ - Eoi$^{280}$). The results in terms of MSLP difference and SLP
SD difference are included in the Supplement material Figures S10 and S11. The MSLP difference (Figure S10) is very
consistent in response to a CO$_2$ doubling (i.e. no substantial change) as well as in response to mid-Pliocene conditions (i.e.
a substantial increase in NP MSLP), for all simulations. The SLP SD difference across the set of simulations (Figure S11) is
similar, but not entirely the same. The NP SLP SD increase for the CO$_2$ doubling with present-day geography (E$^{560}$ - E$^{280}$)
is not seen in the CO$_2$ at mid-Pliocene conditions (i.e. no substantial change; Eoi$^{560}$ - Eoi$^{280}$). The NP SLP SD decrease
for the mid-Pliocene conditions at low CO$_2$ (Eoi$^{280}$ - E$^{280}$ is also seen at higher CO$_2$ (Eoi$^{560}$ - E$^{560}$), but slightly intensified.
Finally, the response to the main mid-Pliocene simulation in PlioMIP2 (Eoi$^{400}$ - E$^{280}$) seems to be an added combination of the
responses to the mid-Pliocene BCs ánd elevated CO$_2$. The reduction of NP SLP SD is seen, but slightly less than in response
to the Eoi$^{280}$, which would be the scaled effect of the response to the E$^{560}$. Research by Garfinkel et al. (2020) shows that the
stationary wave response to various boundary conditions is not always linear and additive, however. A more in-depth analysis
on the nonlinearity and nonadditivity of the winter variability response is out of the scope of this work.

Secondly, we want to hypothesize which specific change in the boundary conditions of the mid-Pliocene simulation could be mainly responsible for the observed responses. As we have not performed these sensitivity studies with our own model, we will turn to results from studies performed with with other PlioMIP2 models. Some modelling groups performed simulations with either just reduced *land ice cover* (including the reduced GIS), or just changes to the *orography*, which includes the closure of gateways (including the Bering Strait and Canadian Arctic Archipelago), vegetation changes, and minor topography and bathymetry changes. Stepanek et al. (2020) performed these sensitivity studies with climate model COSMOS and find that the annual mean climate response (in terms of surface temperatures and precipitation) is generally more sensitivity to the changes in orography, compared to changes in ice sheets. Specifically in terms of tropical Pacific precipitation, the response to orography is substantial (albeit not the same as for our Eoi$^{280}$ simulation), while the response to reduced ice sheets is insignificant (compared to the E$^{280}$). Employing sensitivity studies with climate model CCSM4-UoT, Chandan and Peltier (2018) show that the northern higher latitudes climate (in terms of annual mean as well as winter mean surface temperatures and sea-ice cover) is also more sensitive to changes in orography, than changes in ice sheets. Also using the CCSM4-UoT simulations, Menemenlis et al. (2021) find that a wintertime stationary wave pattern present over the pre-industrial North Pacific nearly disappears in the mid-Pliocene. They show that this reduced wave train response is present in the simulations with changed orography, as well as the simulation with reduced ice sheets (in almost equal amplitude), but not in simulations with just elevated CO$_2$. Regarding which specific orography change could cause most of the response, Otto-Bliesner et al. (2017) uses CCSM4 simulations and finds that there is a similar response in North Pacific Arctic climate to closing just the Bering Strait or just the Canadian Arctic Archipelago. To summarize, studies with other climate models indicate a large sensitivity of the climate response to changes in orography (including the closed gateways), over the changes in ice sheet cover. However, these results do not automatically translate to the winter variability response, and neither to our specific model used here.

### 4.4 A reflection on model performance

When comparing our E$^{280}$ pre-industrial reference simulation with the CR20 reanalysis (see Figure 1), we find some large biases with regards to the MSLP and total SLP SD. The MSLP gradient in the NA is greatly overestimated. However, the NAO and EA are well reproduced, and the main differences within the set of simulations are not in the NA. In terms of SLP SD, there is an overestimation of the variance over the NP, and an underestimation over the NA, which is a known model bias (Eyring et al., 2021). This affects the AO mode, as well as the differences between the CR20 and E$^{280}$ in terms of correlations between the different modes (Figure 6). While the SLP SD in the NP is overestimated, the NPO and PNA are well reproduced both in terms of spatial pattern and in variance explained by each mode. We are therefore confident in our results regarding shifts in NPO and PNA in our set of simulations.

The choice to evaluate only one model was based on the fact that CCSM4-Utr has a unique set of sensitivity simulations that most other PlioMIP2 modelling groups have not performed. However, a limitation is that the results are biased towards the performance and sensitivities of CCSM4-Utr. While most PlioMIP2 modelling groups have not performed the Eoi$^{280}$ and E$^{560}$ simulations, we can still try to extrapolate the dynamical explanation based on results with the Eoi$^{400}$ simulation that has been performed by all modelling groups. Most PlioMIP2 models show 1) a northward shift of the ITCZ (Pontes et al., 2022), 2) a

westward expansion of the Walker circulation (Han et al., 2021), ánd 3) a reduction of the ENSO variability (Oldeman et al., 2021) in the mid-Pliocene Eoi[400]. Considering our dynamical hypothesis linking shifts in Pacific convection to the changes in SLP variability, this could indicate that we can expect similar changes as found in CCSM4-Utr across the PlioMIP2 ensemble. However, for all of these three features, the CCSM4-Utr Eoi[400] simulation is the model with the largest change in that direction, which would imply that CCSM4-Utr might be an end-member of the spectrum. An assessment of the changes in NP variability and their relation with ENSO changes in the full PlioMIP2 is planned for future research.

Haywood et al. (2020) assess climate sensitivity across the PlioMIP2 ensemble, and report that CCSM4-Utr has a relatively low equilibrium climate sensitivity (ECS; 3.2 compared to 3.7 in the ensemble mean). However, it has a relatively high Earth system sensitivity (ESS; 9.1 compared to 6.2 in the ensemble mean). ECS and ESS are both measures of the climate's sensitivity to a $CO_2$ doubling, but ESS takes into account long term feedbacks such as adjustment to changes in ice sheets and orography, whereas ECS just considers $CO_2$ sensitivity. CCSM4-Utr's high ESS, but low ECS, indicates a large sensitivity to the mid-Pliocene boundary conditions other than elevated $CO_2$. This would indicate that while the response in the Eoi[280] simulation might largely dictate the response in the Eoi[400] for CCSM4-Utr, this does not have to be the case for many of the other PlioMIP2 models.

## 4.5 The mid-Pliocene as future climate analogue?

In the introduction, we posed the question of whether the mid-Pliocene can be considered as an analogue for future changes in atmospheric winter variability in the NH. In terms of temperature and precipitation, Burke et al. (2018) show that end of this century climate would be most analogous to the mid-Pliocene, both following the RCP4.5 and RCP8.5 scenarios, compared to other past warm climates. We also find that the winter SAT response to elevated $CO_2$ or to mid-Pliocene boundary conditions other than $CO_2$ are similar. However, we find that the response in winter SLP variability is almost opposite to both forcings, especially over the North Pacific. This is similar for the mid-Pliocene ENSO and AMOC, where the changes are also opposite to end-of-the century projections (Pontes et al., 2022; Weiffenbach et al., 2023). This could already disqualify the mid-Pliocene as a future analog for these climate features, but one could argue that some mid-Pliocene boundary conditions are actually long-term Earth system feedbacks to elevated atmospheric $CO_2$, such as the changes in ice sheets and vegetation (Feng et al., 2022). However, following the sensitivity study results in Chandan and Peltier (2018), it seems that the biggest influence on mid-Pliocene higher latitude climate are changes in orography, which includes the closed Arctic gateways, rather than the reduced Greenland Ice Sheet, or elevated $CO_2$. The Bering Strait is not expected to close in the coming centuries or millennia, so this disqualifies the mid-Pliocene as analogue for a future warm climate. However, the analysis in this paper is restricted explicitly to the NH winter. Other seasons and regions in the mid-Pliocene climate may behave differently, and potentially more analogous to what is expected for increased future $CO_2$ levels. Furthermore, other models might be less sensitive to the mid-Pliocene boundary conditions, and could be better candidates to assess future climate analogy.

## 5 Summary and conclusions

In this study, we address the question whether the mid-Pliocene climate be used to investigate the response of NH winter atmospheric variability, such as the NAO, NAM and PNA, to increased $CO_2$. To answer that question, we want to know 1) whether there is a difference in the response to elevated $CO_2$, and to mid-Pliocene boundary conditions other than $CO_2$, including closed Arctic gateways and reduced ice sheets, and 2) how changes in mean winter climate relate to changes in atmospheric variability in the NH. We use a version of the CESM1.0.5, that is a part of PlioMIP2, and we show that the pre-industrial reference simulation is good in reproducing patterns of SLP variability, when compared to the CR20 reanalysis. We use a set of sensitivity simulations to separate the response to a $CO_2$ doubling and the response to mid-Pliocene boundary conditions other than $CO_2$.

Considering subquestion 1), we find that, although winter surface temperatures respond in a similar way to $CO_2$ or to mid-Pliocene boundary conditions, the responses in mean SLP, jet stream and precipitation show distinct differences. In the $CO_2$ doubling simulation, there is little response in mean SLP, while the mid-Pliocene shows a large increase in mean SLP over the North Pacific, together with a weakened jet stream. Regarding SLP variability, we find an increase in North Pacific SLP variance and a strengthening of the Pacific-North American pattern (PNA) in response to $CO_2$ doubling, consistent with literature, and no notable changes over the North Atlantic. An opposite response is seen in the mid-Pliocene boundary conditions simulation, where SLP variance decreases over the North Pacific, the PNA becomes weaker, and the North Pacific Oscillation (NPO) becomes the dominant mode of variability. We find a strong correlation between the PNA and North Pacific jet intensity on the one hand, and the NPO and jet latitude on the other hand, both in pre-industrial and mid-Pliocene climate.

Considering subquestion 2), we find that the pre-industrial climate, characterised by a strong PNA, shows a strong mean jet with variations in jet strength but not in latitude. The mid-Pliocene boundary conditions simulation shows a weak mean jet that is less variable in intensity, but has a high level of variation in jet latitude, consistent with a dominant NPO. We hypothesize that the weakening of the North Pacific jet stream is ultimately related to shifts in the tropical Pacific precipitation signal, both in the mean and in its variability. Both a reduced ENSO variability, as well as shifts in the mean ITCZ position and upward branch of the Walker circulation, weaken the tropical Pacific precipitation response, which in turn leads to a weaker North Pacific jet stream.

The large sensitivity of the mid-Pliocene climate to changes in boundary conditions other than elevated $CO_2$ leads us to conclude that the mid-Pliocene climate might not be a good candidate to determine the response of NH winter atmospheric variability to increased $CO_2$. The opposite response in North Pacific winter variability to elevated $CO_2$ as opposed to mid-Pliocene boundary conditions weakens the notion of the mid-Pliocene climate as a future analogue, especially since the closure of the Arctic gateways is not a long-term Earth system feedback to elevated $CO_2$.

Even though we consider the CESM1.0.5 a suitable model for this study, we think that it might be an end-member in the PlioMIP2 ensemble in terms of its winter variability response. In future research we plan to extend the investigation to the rest of the PlioMIP2 ensemble. In addition, the specific mechanisms by which the different mid-Pliocene boundary condition cause the changes in SLP variability and jet variations can be investigated in more detail. The specific response to these individual

forcings, and the possible non-linear interaction between them, is an important open issue, which if resolved can help us understand how the winter climate responds to changing forcings in a warmer future.

*Code availability.* The Python scripts (Jupyter Notebooks) used for data analysis are freely accessible through Zenodo (https://doi.org/10.5281/zenodo.782
Oldeman (2023))

*Data availability.* Output from CCSM4-Utr simulations is available upon request. Please contact Michiel Baatsen (m.l.j.baatsen@uu.nl) for access. NOAA/CIRES/DOE 20th Century Reanalysis (V3) data is provided by the NOAA PSL, Boulder, Colorado, USA, from their website at https://psl.noaa.gov (last access: 28 February 2023). An evaluation of the performance of the CR20v3 can be found in Slivinski et al. (2021). Support for the Twentieth Century Reanalysis Project version 3 dataset is provided by the U.S. Department of Energy, Office of
Science Biological and Environmental Research (BER), by the National Oceanic and Atmospheric Administration Climate Program Office, and by the NOAA Earth System Research Laboratory Physical Sciences Laboratory.

*Author contributions.* All authors contributed to the design of this work. AMO performed the analyses and wrote the manuscript.

*Competing interests.* The authors declare no competing interests.

*Acknowledgements.* The work was carried out under the program of the Netherlands Earth System Science Centre (NESSC), financially sup-
ported by the Ministry of Education, Culture and Science (OCW grant number 024.002.001). Simulations were performed at the SURFsara Dutch national computing facilities and were sponsored by NWO-EW (Netherlands Organisation for Scientific Research, Exact Sciences) (project nos. 17189 and 2020.022). The authors would like to thank Michael Kliphuis for setting up and managing the CCSM4-Utr model simulations, and Ezekiel Djeribi Stevens for their performed analysis and interpretation on the CCSM4-Utr NAO. The authors express their gratitude to two anonymous reviewers and the co-editor for providing useful feedback on an earlier version of the manuscript.

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
