# Peer review of "Mid-Pliocene not analogous to high CO2 climate when considering Northern Hemisphere winter variability"

_EGUsphere, 2023_

## Editor Comment (EC1)

This paper presents EOFs of January NH midlatitude circulation variability in the Pliocene from the output of a set of previously published experiments from one model used in the PlioMIP2 project. The authors conclude the Pliocene climate is not an analog for a future climate under increasing $CO_2$ because the variability in NH January is different because of differences in boundary conditions (orography) during the Pliocene vs. modern, underlining the results from Menemenlis et al. (2021) who showed the Northern Hemisphere stationary wave is greatly reduced in the same model when late-Pliocene boundary conditions are used in place of modern day boundary conditions.

1. The manuscript falls short, however, in providing a dynamical analysis of why the variability (and the mean state) changes under Pliocene boundary conditions. Further analysis should be done to demonstrate why the variance in the various patterns changes. For example, why does the variance in the PNA change? Is it due to a reduction in the mean state stationary wave – the main source of energy for the PNA (see, e.g., the discussion on page 237 of Wallace et al. 2023) – but a dynamical analysis should be performed to confirm this. Or is it due to a reduction in ENSO variability? The authors should quantify the different contributions to the change in variance of the PNA. Similarly, evidence through analysis should be provided on why the variance in the NPO changes.

2. The discussion of why the surface air temperature changes in response to changing boundary conditions is speculative: without a quantitative analysis of the thermodynamic energy budget, one can't discern the relative importance of changes in the mean state stationary wave vs rectified effects of changes in the (PNA) transients. The changes in the mean state circulation would probably create a pattern of warming/cooling in the N. Pacific that is very similar to that in Fig. 3c, but it isn't clear to me that this pattern could result from changes in the variability in the PNA (as is argued in section 4.2). To support this claim, the revised paper should show the rectified effect of changing PNA variance and a quantitative analysis of the relative contributions of the mean state and transient changes to the thermodynamic balance warming tendency (e.g., calculate the changes in $\delta(\nabla \cdot \overline{\boldsymbol{v}'T'})$, $\delta(\nabla \cdot \overline{\boldsymbol{v}}\,\overline{T})$, $(\nabla \cdot \overline{\boldsymbol{Q}})$, etc, where the overbar denotes time mean and the prime denotes transients).

3. Concerning the changes in the mean state, Menemenlis et al. (2021) documented that the Northern Hemisphere stationary wave is greatly reduced in this model when late-Pliocene boundary conditions are used in place of modern day boundary conditions. Here, the authors speculate (using the results in section 4.2 of Hurwitz et al) that SLP increases in the Aleutian low in the Pliocene because of increases in SST in the N. Pacific. It is difficult to say for sure (because of the lack of contours and/or the poor resolution in the color bar used in Fig. 3c and other figures) but I don't think the scaling works. Hurwitz et al show a 30 m geopotential response at 850 hPa for a 2C warming in the N. Pacific, which amounts to approximate 3 hPa SLP response (=30 m *(hPa / 8 m) * 850/1000 ) for a 2C anomaly, or 1.5 hPa per 1 C anomaly. In response to Pliocene orography, there is a 16 hPa increase in the Aleutian Low and a ~4 C increase in N. Pacific SST (Figs. 2c and 3c), which is almost three

times greater than the response to the prescribed SST anomalies. Indeed, the SST anomalies seem to be a response to the changes in the stationary wave, not the other way around.

4. Another, more likely, cause of the stationary wave response to Pliocene boundary conditions is changes in the tropical Pacific diabatic heating (precipitation). There is a long literature dating back to Simmons et al. (1983) that shows the strength of the Aleutian low and the amplitude of the stationary wave is sensitive to small changes in diabatic heating over the Maritime continent. Figure 1 of Menemenlis et al. (2021) shows that, in response to Pliocene boundary condition, precipitation is reduced over the far western Pacific and increased in the (unrealistic) double ITCZ in central and eastern Pacific. Hence, it would seem changes in the tropical Pacific climatology could easily be responsible for the changes in the climatological mean state Aleutian Low and the stationary wave (at least, in the Pacific), for the weakening and broadening of the climatological mean jet, and for the changes in the variability in the PNA. Simple AMIP experiments using prescribed climatological SSTs taken from the E280 and Eoi280 simulations would illuminate the cause(s) for these changes in the simulations.

5. January and February are special months in the N. Pacific when the jet takes on a more subtropical location and becomes strong and supports less variability – the so-called Pacific mid-winter suppression of the jet. I am not surprised that the EOFs of DJF circulation change in a similar in the Pliocene to those shown in the paper for January (but showing that analysis instead of the analysis of January only would boost the statistical significance of the results). Perhaps even more interesting, it is less clear the other winter months – ONDM, the stormy months in the Pacific – will show the same Pliocene minus modern differences as those in the mid-winter suppression months. Streamlining the introduction and discussion of previous results concerning mean state changes and removing tangential discussion on changes in heat transport in section 4.2 would leave room for a comparison of the changes in variability.

6. Consider analyzing the variability and mean state changes in at least one other climate model used in the PlioMIP2 project. Are your results sensitive to the model used? Fig 1a of the paper shows that the biases in the modern day January stationary wave in the model are large – about twice too large in the N. Atlantic and 40% too large in the N. Pacific – and so too is the variability too large – by a factor of 2 or three.

7. The use of nonstandard (and apparently arbitrary) assignments of the labels "zonal" and "azonal" terminology to describe well know patterns of atmospheric variability is needlessly confusing. Without further justification, I strongly urge the authors to use standard monikers for these patterns to avoid needlessly confusing the readers. [E.g., the NAO and NPO describe regional-scale patterns of variability featuring meridional dipoles in geopotential, changes in the jet strength, and changes in the meridional location of the storm track. It is difficult to see how that fits with the monikers "zonal" and "azonal".]

8. Consider using ERA5 instead of CR20 for the modern "observations", or truncate the CR20 period to start in the early-mid 1900s. The former has 72 years of very good data; the latter is less constrained – especially in the first half of the analysis period used (1836-2015).

9. I agree with both reviewers that the title doesn't fit the contents of the paper (e.g., the title refers to generic warm climates rather than the late Pliocene) and that adding an analysis of the response to an increase in CO2 under late Pliocene conditions (the change in the pair of experiments Eoi400 and Eoi280) would add new results to the paper (*vis à vis* the response to increased CO2 under different boundary conditions).

References not already included in the manuscript

Simmons, A. J., J. M. Wallace, and G. W. Branstator, 1983: Barotropic wave propagation and instability, and atmospheric teleconnection patterns. *Journal of the Atmospheric Sciences*, **40**, 1363–1392.

Wallace, J. M., D. S. Battisti, D. W. J. Thompson, and D. L. Hartmann, 2023: *The Atmospheric General Circulation*. Cambridge University Press, 424 pp.

---

## Author Comment (AC1)

Answers to: RC1

https://doi.org/10.5194/egusphere-2023-757-RC1

Dear anonymous reviewer 1,

First of all, many thanks for considering our manuscript for publication and taking the time to review. We value that you think our work is worthy of publication after revision and responding to your feedback. We appreciate your positive words, and we acknowledge your critiques and constructive feedback. In this document we will aim to answer your questions and respond to your feedback, in a way that we hope to be satisfying. Ultimately, we think the manuscript will be improved and will be accepted for publication after intended revisions.

General comments.

This paper explores mean state and jet variability changes in response to increased CO2 and mid-Pliocene boundary conditions through previously published general circulation model experiments using the CESM. The experiments are well-designed, and some interesting, although perhaps not too surprising, results are found, that changes in climate variability can be very different in response to increased CO2 relative to other boundary condition forcings that also create warmer climates. Overall, the structure of the manuscript is logical, and figures are clear. However, I find much of the results section to be very descriptive of the figures, with little interpretation, leaving the reader to try to find their own interpretation - more interpretation of the results (and less description) would greatly improve this manuscript.

We agree the Results section is quite descriptive and could use more interpretation. This was a choice we made, but we understand it doesn't read well and leaves interpretation to the reader, which is not the intent. We propose the following:

- We will reduce the level of description in the Results, in those places where the description is not really necessary for answering the research questions, or where description is redundant because it is obvious from the presented Figures. Example 1: L263 – 265 ("Temperatures increase … higher latitudes.") these two sentences will be combined and thus shortened, as it is merely description. Example 2: L289-290 ("The dynamical … Eurasian continent.") this sentence will be removed as it is again merely a description of the figure with no interpretation.
- We will increase interpretation in the Results in two ways.
    o First is to move some of the interpretation that is currently presented in the Discussion, specifically Section 4.2 Physical and dynamical interpretation, to the parts in the Results where it would fit. For example L438-445, a paragraph offering interpretation of the CO2 doubling results and making a link between the mean changes and variability changes, can almost entirely

be moved to the Results section 3.2.1 (Sea-level pressure variability / CO2 doubling).

- Secondly we intend to include subquestions to the main research question in the Introduction (including: "How do changes in the mean winter state relate to changes in winter atmospheric variability?") as well as some hypothesis based on literature that is now newly presented in the Discussion (mainly section 4.1). In the Results section, when presenting our results we can then refer back to the questions posed in the introduction, as well as to our initial hypothesis. For example, linking mean state changes to changing in variability is what we are doing in the previously mentioned paragraph L438-445. Moving that to the Results, and referring to the question asked, will hopefully increase the level of readability of our Results section. (more on rewriting/reordering of the Introduction below)

The discussion section leans heavily on prior literature and could do a much better job of putting the new results in the context of literature and highlighting the novel results in this study and their relevance.

We agree that our Discussion is currently introducing some literature that was not treated earlier in the paper (for example, in the Introduction). Next to that, how the Discussion is currently written can indeed be confusing as to when we refer to our own results, and when to previously published results. We propose the following:

- We intend to reorder the Introduction (more on that below), where we will introduce a paragraph on the effects of the different boundary conditions of the Pliocene on the climate. This would include parts of section 4.1 Sensitivity to mid-Pliocene boundary conditions. For example paragraph L417-424 is treating SAT and SIE response to different mid-Pliocene boundary conditions in other climate models, which would actually be relevant to mention in the Introduction.

Including uncertainty estimates on some of the quantities reported would also improve the paper.

Thank you, we will take that into consideration. Your suggestion to calculate uncertainties of the changes in variance explained is one thing we will do. We furthermore plan to include statistical significance information on the correlation coefficients calculated, for example in Figure 7 (where some correlations might not be statistically significant), and Figure 8.

The research is motivated by the question of whether "the mid-Pliocene climate be used to assess the response of present-day Northern Hemisphere winter atmospheric variability, such as the NAO, NAM and PNA, to increased CO2"; however, I remain a little unclear as to why this is a valuable question to ask? If we trust the model in reproducing climate variability of the mid-Pliocene, why don't we just trust the models for the future? One argument for using paleo data is that we have proxy observations and so don't need to rely only on models, but here you are just using a model, so any model biases remain. Exploring and understanding climate variability in different climates is interesting and useful for

understanding the underlying climate system and behaviour of internal variability, and your result that variability is very different in two different warm climates is interesting. These insights could be helpful, in a more indirect way, in helping understand projected future changes. I think the research focus is interesting and worthy of publication, but the motivation does not convince me as it is currently presented in the introduction, and I found the structure and order of the introduction rather confusing.

For example, the paragraph starting on line 42: "An issue with investigating the response of climate variability to increasing CO2 in the near-future is that the present-day climate system is not in equilibrium with the mean forcing…." – are you saying that the present day isn't an analog for the future because we're not in equilibrium? Or that, for model simulations in which the climate is changing, it is difficult to assess what is trend and what is a change in variability? I would argue that, at least in the near-future, the climate won't be in equilibrium, so the modelled simulations of future climate seem like a better analog than a past climate that is in equilibrium? Even if you argue that you can only study this in an equilibrium climate, why not use simulations of future changes that have reached equilibrium, e.g. 4xCO2?

We acknowledge your critical feedback on the formulation of the motivation as well as the structure of the introduction. We believe our motivation is valid, but can understand that how it is currently phrased and presented (which includes the research question and structure of the introduction) can be confusing, or not convincing. Instead of trying to answer all the question you asked above, we propose a different structuring of the Introduction and a more clear formulation of the motivation and research questions. We believe that will cover most and hopefully all criticisms you pose.

Proposal of paragraphs of the Introduction:

1. Introduce the midPliocene as most recent period with atmospheric CO2 similar to present-day. Explain why midPliocene is considered a possible 'best analog' for near-future climate (when considering mean temperature and precipitation and compared to RCP4.5 projections, see Burke et al 2018) – would include parts of current L42-51
2. Introduce midPliocene modelling efforts – would include current L52-60
3. Explain midPliocene can also be relevant for long term climate projections, in terms of CO2 (400ppm similar to optimistic SSP1-2.6 scenario in 2300), or when considering reduced Greenland ice sheet and West Antarctic ice sheet. – would include some parts and literature that is now presented in Discussion 4.4 The mid-Pliocene as future analog?
4. Explain next to CO2 and reduced GIS, there were also closed gateways, which previous studies have shown to have effect on mean climate – move some literature and sentences from Discussion section 4.1 Sensitivity to mid-Pliocene boundary conditions here
5. Apart from mean climate, we can also study climate variability in the past, in order to further understand climate dynamics & variability response in warm climates, as well as how mean climate changes relate to changes in variability - Cite work on midPliocene ENSO, a.o.

6. Also very relevant to study winter variability, such as NAO. Large impacts, but future projections not in agreement. What happens with winter variability in warmer climates? – This would largely be L19-41
7. Brings us to the rephrased research question: "Can the mid-Pliocene be used to investigate the response of NH winter variability to a warmer climate?" with subquestions: 1. Is there a difference in the response to elevated CO2, and to mid-Pliocene boundary conditions other than CO2, including closed Arctic gateways and reduced ice sheets? And 2. How do changes in mean winter climate relate to changes in atmospheric variability?
8. Introduce the model used and the specific simulations used to answer the questions – corresponding to L87-L95
9. Briefly hypothesize what we expect the answers to be based on previous studies – this would consist of parts from L61-L76, section 4.1, and section 4.3
10. Paper outline – current L96-100

In the current Introduction, the focus might be too much on using the midPliocene to 'fix' the fact that current climate projections on winter variability are uncertain or not consistent. From that train of thought, we can understand your comment "why don't we just trust the models for the future". Hopefully, with the newly proposed structure of the Introduction, it is clearer that we study the midPliocene because it can help us to understand warm, high CO2 climates, and we investigate winter variability in that climate to get a better grasp of winter variability in warm climates.

Lastly, the title of the paper suggests this is a response that is consistent across many past warm climate conditions, rather than just the mid-Pliocene – given the suggested dependence on orography, and possible dependence on SSTs, this may not be true;  I suggest to be more specific in the title so it's not potentially misleading.

Thank you, we agree the title can be potentially misleading. The other reviewer also has a comment on the title, so to answer to both, we propose the following title:

**"Mid-Pliocene not analoguous to high CO$_2$ climate when considering Northern Hemisphere winter variability"**

Specific comments

'the geological past climate was in equilibrium with forcing' – if this was always true, the climate couldn't have changed in the past? You could argue that it was, most of the time, more in equilibrium than we are today.

L44. Indeed, depending on the timescale considered, the climate was not always in equilibrium, as it has adapted to changing ice sheets, atmospheric CO2 levels, etc. To clarify, we will add a timescale, for example ".. on timescales relevant for climate variability..".

You mention variability on decadal timescales, but not the PDO (Pacific Decadal Oscillation), which surprised me – is there a reason to not discuss the PDO in modes of variability of the

North Pacific? Particularly given, on line 175, you say you are mainly interested in interannual and decadal variability.

The reason we did not mention the PDO in the Introduction is that consider atmospheric variability, and the PDO is classically defined as an oceanic mode of variability. However, it is true that the PDO has a clear connection to the atmosphere, with the PNA as the most obvious teleconnection (e.g. Chen et al 2018). In that light, it is relevant to mention the PDO earlier in the manuscript. We will include a sentence in the Introduction that mentions the PDO as well as ENSO and their deterministic links to NH atmospheric variability.

Line 173. What re-analysis data do you use? Do you have enough years to use a 50-year window?

We compare with NOAA's CR20 reanalysis, as is stated in section 2.3 Validation of E280 with reanalysis data. We propose to remove the mention of reanalysis in L173 and move the explanation that we apply Lowess filtering to remove climate change trends to section 2.3

Line 175 'A window size of 50 years was chosen since we are mainly interested in interannual and decadal winter variability' what degree did you use for your Lowess smoothing, and aren't you removing all of the interannual variability and most of the decadal variability by using a 50 year window?

- *Check Lowess degree*
- We understand the confusing regarding the Lowess filtering; we **filter** the data by **removing** a Lowess smoothing with a 50y window. So we are left with anomaly data where all the variability with periods larger than 50years are removed, and the periods below 50y are still present. We propose to rephrase: "Before analysis, a Lowess smoothing using a 50 year moving window is removed from anomalies at each spatial grid point."

Line 185: how do you determine level of zonality? Just subjectively by looking at the EOFs or some objective method?

We determine level of zonality by subjectively looking. We have employed a method earlier that used the same definitions used for changing sign of the EOF (L188-190), but this still required a lot of (subjective) tuning so that it worked well. In the end, we are just dealing with two times four EOFs per basin (CR20, E280, Eoi280, E560), so we chose to determine the level of zonality qualitatively.

Furthermore, we propose to stick to the known nomenclature (NAO, PNA, etc) instead of NAtl-z, NPac-a, in order to answer comments from Reviewer nr2. We think that the current naming convention, including the definition of 'zonality', might confuse the reader. Sticking to known nomenclature will help, with the side note that the modes in the midPliocene might not be exactly what we expect from the present-day. This is for example also what has been done in previous studies regarding NAO in past climate (e.g. the LGM, see https://journals.ametsoc.org/view/journals/clim/23/11/2010jcli3372.1.xml)

Figure 1 caption. I think d) should be NPac-z not NPac-a?

Correct. We will change this in NPO.

Figure 1. It's a little confusing whether the contours are the absolute CR20 values, or the differences between E280 and CR20. I think perhaps it is differences for a. and b. and then absolute values for the others, but the caption does not make this clear.

It does specify it currently in the caption: "For (a) and (b), the contours represent the difference between the E$_{280}$ and CR20, while (c)-(g) show the CR20 EOFs in contours." To clarify, we can add ".. show the **absolute values of the** CR20 EOFs …"

Line 198. MAX and MIN would seem more related than PLUS and MIN

It was chosen to stick with nomenclature that is used in literature, at least commonly with the NAO phases (NAO+ and NAO-). Since we plan to adopt NPO instead of NPac-z, we will also change this to call the phases NPO+ and NPO- instead of NPac-z PLUS and NPac-z MIN. We hope that will be clearer to the reader.

Section 2.3. The description of the re-analysis dataset would be better earlier, before you mention that you've used re-analysis data in line 173.

We decide to remove the mention of the reanalysis in L173 (see earlier answer).

The way you have described the re-analysis data is confusing as 'we use assimilated sea-level pressure data from the NOAA 20C reanalysis….' – this almost sounds like you've done some post-processing to the re-analysis data, or that you're using the data that was assimilated into the re-analysis. 'The data runs from 1836 to 2015 and is assimilated using surface pressure observations on a 1.0$^\circ$ latitude x 1.0$^\circ$ longitude grid' This sounds like the surface pressure observations are on the 1x1 degree grid, and assimilated the re-analysis, rather than the re-analysis data assimilating the surface pressure observations.

We understand the confusion regarding the current phrasing. We propose to rephrase it as such: "We use **monthly mean sea-level pressure data** from the NOAA … as CR20). The **reanalysis covers the period from** from 1836 to 2015 and **assimilates surface pressure observations in combination with a forecasting model to estimate a set of atmospheric variables.** An evaluation …. Slivinski et al. **The CR20 data is presented on a 1deg latitude x 1deg longitude grid, and** we interpolate the …. Between the E280 and CR20."

I think it is worth mentioning that a benefit of the CR20 for these longer time periods is that there is more consistency in the data that is assimilated than for, for example, ERA5, which includes satellite data in more recent periods. That said, the amount of data being assimilated certainly does change with tie in CR20.

Agreed, and the length of the dataset, plus the fact that it covers more of what can be considered 'pre-industrial', was the key reason for us to choose this reanalysis over ERA5 and ERA20C. We can add a sentence in paragraph L201-207 to clarify: "The length of the CR20 dataset and the fact that it covers a period closer to what can be considered pre-industrial motivated us to use this reanalysis data over more recent reanalyses such as ERA5."

Section 2.3. Why do you use all of the data in the re-analysis, through to 2015, when your simulation is just pre-industrial, as you mention? Do you de-trend the re-analysis data to try to take out any climate change signal in MSLP? Would choosing a shorter time period that is closer to your pre-industrial modelled dataset not be better?

See also the previous answer. Yes, we detrend the data, as is mentioned in 173-174. We will remove that part and move that mention of the detrending here.

Line 209. Do you mean spatial mean MSLP? Is the model bias in global mean sea level pressure particularly meaningful? Seems reasonable this is relatively small since sea level pressure is essentially a measure of the amount of atmosphere above a point, so the global mean MSLP is broadly a measure of how much atmosphere there is in your model?

Yes, we mean spatial mean MSLP. Indeed, it might not be particularly meaningful, although it is not global mean MSLP but NH winter mean MSLP. There could be a discrepancy in atmospheric mass distribution over the hemispheres in a particular season. However, your critique is valid, and we propose to remove this sentence altogether, and instead compute RMSE (computed per grid point, then averaged).

Figure 2. It might be interesting to show Eoi400-Eoi280 as an addition panel in fig 2. I realize this isn't a doubling, and so isn't equivalent to panel b, but it would be interesting to see if the pattern response to increased CO2 is similar in the Pliocene (or you could scale the responses, e.g. to 1 degree of global warming)? This would also help with seeing your result described on lines 253-254, as it seems this result (that the MSLP difference are predominantly caused by the different surface boundary conditions, not the different CO2 levels) is quite a key one, given that one of your main conclusions is about the change in variability in response to these two forcings. However, from the panels presented in Fig. 2 and Fig. S2, it is not easy to understand what comparison to make to come to this conclusion. I think adding Eoi400-Eoi280 to Fig 2 would help.

Thank you for this suggestion. We have actually performed a 'CO2 doubling' experiment in the mid-Pliocene, the Eoi560 (see Baatsen et al 2022). The results of this simulation can be used to compare to the CO2 doubling with pre-industrial BCs (i.e. Eoi560-Eoi280 versus E560-E280). In addition, it can be used to compare to the mid-Pliocene BC effects at high CO2 (i.e. Eoi560-E560 versus Eoi280-E280). We chose not to include the results of this simulation, as we figured it would not be necessary to answer the research questions. However, considering your point here, as well as your later comment on assessing nonlinearity, we propose to include some Eoi560 results in the revised manuscript.

We are not convinced to include the Eoi400 – E280 results in Figure 2. We think it might distract from the results discussed: MSLP response to the different boundary conditions/CO2. We propose to discuss the Eoi400 vs E280 results in more detail in the Discussion, in the section where we currently discuss nonlinearity (L499-506). Specifically, we propose to show a set of figures in the Supplement that we will then refer to, namely the MSLP difference and SLP SD difference for the following five combination (i.e. 5 panels per variable):

1. E560 – E280 (effect of CO2 doubling)
2. Eoi560 – Eoi280 (effect of CO2 doubling with mid-Pliocene BCs)
3. Eoi280 – E280 (effect of mid-Pliocene BCs)
4. Eoi560 – E560 (effect of mid-Pliocene BCs at high CO2)
5. Eoi400 – E280 ('true' mid-Pliocene conditions vs pre-industrial reference)

With this information, we believe that we can properly address nonlinearity, as well as the combined effects of CO2 and other mid-Pliocene BCs in the 'true' mid-Pliocene Eoi400 simulations.

Figure 2. Is any of the signal over Greenland in panel b likely to be because of differences in the height of the ice sheet, and how the interpolation to sea level is performed?

In panel b not, because the GIS is the same in E280 and E560. In panel c (Eoi280 – E280) it could be, since the GIS is different in both simulations. The difference in height, and as such the interpolation of pressure to sea-level, might explain the slightly 'spotty' differences over Greenland. We don't think it is necessary to mention this in the manuscript.

Fig. 3. In understanding the sea ice changes, it seems there are significant changes in land-ocean boundaries (as shown in the coastlines in panel c, which is different to that for present day) but this doesn't seem to be discussed. For example, there looks like there is large sea ice retreat over Hudson's bay, but this looks to be more related to shifting coastlines? Mentioning this would be useful.

You are correct, some of the sea-ice changes in the Eoi280 are due to coastline changes, for example over the Hudson Bay area, but also the Bering Strait. We will make a mention of this in the revised manuscript: "The Eoi280 shows sea-ice retreat over areas that are seas in the present-day geography, but were land in the mid-Pliocene, for example in the Bering strait and Canadian Arctic Archipelago."

Section 3.1. The results are useful for putting the variability differences in context of mean climate changes, but this isn't mentioned explicitly in the text, so the reader is left to add this interpretation themselves. Also, the section is very descriptive, with little attempt to provide mechanistic or physical explanations for the differences you see. For example, can you give some suggestions as to why you see such a strong response in the upper level circulation in response to Pliocene surface boundary conditions? Do you think this is related to differences in topography? Or tropical SSTs? Or something else?

As mentioned before, we will explicitly add the subquestion "How do changes in mean winter climate relate to changes in atmospheric variability?", to place the mean winter results in context of answering the research question. Next to that, as explained before, we will move some of the interpretation that is currently presented in the Discussion to the Results section. In the current Discussion, specifically section 4.2, we already aim to answer some of the questions you pose here.

Fig. 5. Having the positive contours dashed and the negative dot-dashed is a rather subtle difference (although after looking at Fig. 6 I now realize they are also coloured – stating this

in the caption would help). I would find it easier to interpret the figures is positive was solid contours and negative dashed (and zero bold if you wanted to distinguish the zero contour). At least mention whatever convention you use in the caption to make it easy to get the information.

We use negative contours dot-dashed and positive dashed throughout all the figures. We have chosen to use a solid white zero line in the difference contours (fig5,6 a) but dashed black zero line in the EOF figures (fig5,6 others) for the following reasons: solid white is well visible in the difference plot because of the colormap used. In the EOF plots however, the colormap is diverging around white, so a white line to indicate the zero contour is almost not visible. A solid black line was confusing because of the coastlines. This is why we chose to use black dashed for the EOF figures. We will mention the convection in the captions where necessary.

Changes in variance explained – I think you should calculate uncertainties on these values (e.g. using bootstrap resampling), to get a sense of how robust the differences you are reporting are. With 200 months I would hypothesise that changes of 10% are robust, but it depends on low frequency variability in the model. But are differences of 63.9 vs 66.7, or 17.5 vs 20.3 robust, or just noise, I'm unsure.

Thank you for this useful suggestion. We are not sure if bootstrap resampling is the best choice, but we will introduce a method that can help us assess how robust changes in percentage of variance explained are. For example, by computing the EOFs and perc of var explained for every possible 50year sample in the 200year dataset, and using that to construct a 5-95% confidence interval, or a standard deviation.

Section 3.2.3. I am not sure what to conclude from the section – more context is required. I also wonder whether the correlation between the NHem leading mode and the zonal or azonal modes is sensitive to which mode is picked up as the leading mode in Nhem and therefore how meaningful it is in a physical sense? Is the second leading mode of Nhem in Eoi280 more similar to the leading mode in E560 and E280, and therefore would correlated better with NPac-a?

The aim of this section is to assess how the level of (tele)connection between the different modes in the different basins changes. We believe this is relevant information in order to answer to research questions. We also wondered if the NHem mode is not just a construct of the leading modes in the N Atlantic and N Pacific. Considering the spatial patterns (Fig 5,6), this seems to be the case. However, the correlation coefficients presented in figure 7 show the level with which the modes correlate in space (i.e. how similar their principal components are). Qualitatively assessing similarity of the spatial pattern is not enough to know this. For example, in the E560 (fig7b), the NAtl-a mode shows a stronger (anti)correlation with the NHem (-0.57), than the NAtl-z (+0.43). However, from fig 5 you would not so easily see that the NAtl-a correlates more strongly with the NHem than the NAtl-z.

Furthermore, in line of your earlier feedback regarding uncertainty estimates, we will check for all the correlation coefficients in fig7 if they are statistically significant (e.g. p-value<0.05).

Line 378. Could you provide more details for why you conclude that the more northern jet state is stronger? NPac-z seems to correspond more to a meridional shift of the jet strength (as implied in line 379, and from Linkin and Nigam 2008). You do mention in line 228 that NPac-z is linked to meridional 'modulation' of the Asian-Pacific jet, but it's a little unclear what this means (meridional shifts? Changes in jet strength?) and the reader is left to work out a lot of these details themselves.

We conclude that the northbound jet is stronger (in comparison to the southbound jet in the same phase, that is) from the fact that our simple method to determine jet latitude (lat of max U200 in this zonal mean slice) picks up the northern jet more often than the southern jet. If the southern jet was the stronger of the two, the scatter plot distribution at negative NPac-z PC values would be skewed towards the 20-30N latitudes. Instead, it is skewed to the 45-60N latitudes. We will clarify in L228 by rephrasing "meridional modulation of the jet" as "meridional displacement of the jet", of "variations in the latitude of the jet".

Line 381. "the correlation between the two NPac modes" to me implies that the 2 modes are correlated to each other (which they can't be, being EOFs), rather than (on second reading) one is correlated with the jet intensity and one with jet latitude.

Indeed, your second reading is correct. However, we realise that the current phrasing is confusing, so we propose to change it as follows: ".. as well as the correlation between the PNA and jet intensity, and the NPO and jet latitude, suggest the following: …"

Line 414. I assumed 'we want to explore' implied that you were going to analyse some more simulations with different boundary conditions, but it seems more just literature review.

We understand this can be confusing. As mentioned before, we will move parts of this section to the Introduction, so that it doesn't read as a literature review anymore. We still think this is a useful section in the Discussion, and although we have not performed any additional sensitivity experiments in order to investigate the difference between closed gateways or a reduced GIS, other have. We think that we can use the results of those studies to put our own results into context. However, we will rephrase it like that at the start of the paragraph, so that it is clear for the reader that we are not introducing any new simulations with CCSM4-Utr here.

Line 415-416. What about sea surface temperature changes? Surely these could impact the mean state, and thus potentially variability as well? Tropical SSTs in particular can have a large impact on extratropical atmospheric dynamics (e.g. https://journals.ametsoc.org/view/journals/clim/14/4/1520-0442_2001_014_0565_tiotss_2.0.co_2.xml ). Indeed, in Chandan and Peltier (2018) they note that their model does not agree with some paleoproxies of SSTs in some tropical

locations. Does your model have the same bias? This is a limitation that should be discussed in this paper.

- Specifically regarding L415-416: SSTs are not boundary conditions but rather an effect of boundary conditions, hence we don't include SSTs in the list of topics in L415-416
- Yes, we expect SSTs to influence atmospheric variability. Not just tropical SSTs but also higher latitude SSTs (see e.g. Hurwitz et al that we included in our references, https://agupubs.onlinelibrary.wiley.com/doi/full/10.1029/2012JD017819) . Thank you also for sharing the Yin & Battisti paper. We will outline the link between pacific SSTs and atmospheric variability briefly in the Introduction (see also our answer to your question regarding the PDO), and return to it in the Discussion. We chose to stick with SAT since we are considering atmospheric dynamics and because it is a good proxy for SSTs as well, especially in the mean patterns.
- The teleconnection with the tropics is relevant to mention, as we briefly do in section 4.3 A climate variability point of view, but we believe it is out of the scope of this study to dive into the details. We are presently preparing another manuscript exploring the mid-Pliocene tropical /ENSO teleconnection with the North Pacific in the full PlioMIP2 ensemble.
- Our model does a reasonable job in representing mid-Pliocene tropical proxies, see e.g. Baatsen et al 2022 Fig 9 https://cp.copernicus.org/articles/18/657/2022/ and discrepancies mostly occur in coastal regions with strong boundary currents, that might not be properly resolved in our model resolution.

In most of your discussion I find it hard to understand which is new information and understanding produced by this paper, and which is taken from the literature – it seems like a lot of discussion of the literature, and perhaps some of this belongs in the introduction?

Thank you for this feedback. We feel we have already addressed this issue earlier in our answering.

Line 499. I'm not sure you really address the non-additivity here, you just discuss the literature saying that the responses might be non-additive? It seems like perhaps you don't have all the experiments you would need to address non-additivity: you'd need E400 to compare the differences to CO2 forcing (E280 vs E400), boundary condition forcing (E280 vs EOI280) relative to both together: E280 vs EOI400). And how are you defining nonlinearity and nonadditivity distinctly?

Thank you for this comment, indeed we are currently just discussing that literature study tells us the responses might be non-additive. We feel this is a relevant point to make. Regarding the nonlinearity, we feel we have addressed this point earlier in our answering, when treating the Eoi560 simulation.

---

## Author Comment (AC2)

Answers to: RC2

https://doi.org/10.5194/egusphere-2023-757-RC2

Review of: Past warm climate conditions show a shift in Northern Hemisphere winter variability towards a dominant North Pacific Oscillation (Oldeman et al., 2023)

Authors: Arthur M. Oldeman, Michiel L.J. Baatsen, Anna S. von der Heydt, Aarnout J. van Delden, & Henk A. Dijkstra

Dear anonymous reviewer 2,

First of all, many thanks for considering our manuscript for publication and taking the time to review. We appreciate your positive words, and we acknowledge your critiques and constructive feedback. In this document we will aim to answer your questions and respond to your feedback, in a way that we hope to be satisfying. Ultimately, we think the manuscript will be improved and will be accepted for publication after intended revisions.

In this study the authors pose the question of whether the mid-Pliocene can act as a suitable analog for future climate change scenarios given the relatively similar $CO_2$ concentrations and geography to present day. In particular they focus on suitability for studies of NH winter variability. They use a suite of simulations from CESM1.0.5, part of PlioMIP2, to separate the climate response to $CO_2$ doubling vs. mid-Pliocene boundary conditions. A major finding of this manuscript is that the dominant mode of North Pacific variability differs depending on whether we consider $CO_2$ doubling or mid-Pliocene boundary conditions: the preindustrial dominant mode (the PNA) strengthens under climate change whereas under mid-Pliocene boundary conditions the NPO becomes the dominant mode. The authors conclude by saying that the mid-Pliocene is not a suitable analog for future climate change.

I thank the authors for an interesting study. While the result that different boundary conditions lead to different climate variability is not necessarily shocking, it's valuable to explain that the North Pacific variability in particular changes depending on the forcing. I have some questions and suggestions that I hope will be addressed before publication. I hope the authors find the commends below constructive and useful and I look forward to reading their responses.

Overarching Comments

1. General Motivation: Overall I found the general motivation of this paper to be somewhat unclear. After requisite background on NH wintertime variability and why the mid-Pliocene is considered a potential analog for future climate change, the paragraph beginning at L61 seems to lay out the main question of this study well. The authors then proceed to cite studies which say the two periods are not good analogs for each other. My question, then, is where does the current study fit in? Do prior studies not discuss wintertime NH variability between the periods? Does this study start from the hypothesis that the two periods will be not agree well but the authors seek to quantify that hypothesis? As you clarify this, the discussion at L417 (and throughout Sec 4.1) might become more suited for the introduction.

Thank you for this remark. We see that the current structure of the Introduction might be confusing, and a different structure and ordering can help to put the results and discussion in better context. We propose to restructure the Introduction (also to answer a main critique by Reviewer1), and introduce subquestions to the main research question, a hypothesis based on previous work (to clarify "why does this study fit in?") and move some literature introduced in Discussion 4.1 to the Introduction.

Proposal of paragraphs of the Introduction (for more context see also the answers to RC1):

1. Introduce the midPliocene as most recent period with atmospheric $CO_2$ similar to present-day. Explain why midPliocene is considered a possible 'best analog' for near-future climate (when considering mean temperature and precipitation and compared to RCP4.5 projections, see Burke et al 2018) – would include parts of current L42-51
2. Introduce midPliocene modelling efforts – would include current L52-60
3. Explain midPliocene can also be relevant for long term climate projections, in terms of $CO_2$ (400ppm similar to optimistic SSP1-2.6 scenario in 2300), or when considering reduced Greenland ice sheet and West Antarctic ice sheet. – would include some parts and literature that is now presented in Discussion 4.4 The mid-Pliocene as future analog?
4. Explain next to $CO_2$ and reduced GIS, there were also closed gateways, which previous studies have shown to have effect on mean climate – move some literature and sentences from Discussion section 4.1 Sensitivity to mid-Pliocene boundary conditions here
5. Apart from mean climate, we can also study climate variability in the past, in order to further understand climate dynamics & variability response in warm climates, as well as how mean climate changes relate to changes in variability - Cite work on midPliocene ENSO, a.o.
6. Also very relevant to study winter variability, such as NAO. Large impacts, but future projections not in agreement. What happens with winter variability in warmer climates? – This would largely be L19-41
7. Brings us to the rephrased research question: "Can the mid-Pliocene be used to investigate the response of NH winter variability to a warmer climate?" with

subquestions: 1. Is there a difference in the response to elevated CO2, and to mid-Pliocene boundary conditions other than CO2, including closed Arctic gateways and reduced ice sheets? And 2. How do changes in mean winter climate relate to changes in atmospheric variability?

8. Introduce the model used and the specific simulations used to answer the questions – corresponding to L87-L95
9. Briefly hypothesize what we expect the answers to be based on previous studies – this would consist of parts from L61-L76, section 4.1, and section 4.3
10. Paper outline – current L96-100

2. Results Section: As it stands, Section 3 reads as if the authors made a list of plots and then simply describe them to the reader. Instead of this, I recommend the authors consider their primary motivation (see above) and then present their results in a way that strengthens their argument and presents a coherent storyline rather than just marching through mean SLP, SST, and U200, then std SLP, SST, and U200, and then figures about the jet. For instance, if the hypothesis is that the two eras won't agree well, weave in interpretation of what you're seeing in the difference between the b and c panels of Figs 2-4. Explain why differences in the EOFs lead you to consider the jet, how the differences in surface forcing impact the jet and weather, etc. Interpretation is not inappropriate in the Results section and will lead to a stronger article.

We agree the Results section is quite descriptive and could use more interpretation. This was a choice we made, but we understand it doesn't read well and leaves interpretation to the reader, which is not the intent. We propose the following:

- We will reduce the level of description in the Results, in those places where the description is not really necessary for answering the research questions, or where description is redundant because it is obvious from the presented Figures.
- We will increase interpretation in the Results in two ways.
  - First is to move some of the interpretation that is currently presented in the Discussion, specifically Section 4.2 Physical and dynamical interpretation, to the parts in the Results where it would fit. For example L438-445, a paragraph offering interpretation of the CO2 doubling results and making a link between the mean changes and variability changes, can almost entirely be moved to the Results section 3.2.1 (Sea-level pressure variability / CO2 doubling).
  - Secondly we intend to include subquestions to the main research question in the Introduction (including: "How do changes in the mean winter state relate to changes in winter atmospheric variability?") as well as some hypothesis based on literature that is now newly presented in the Discussion (mainly section 4.1). In the Results section, when presenting our results we can then refer back to the questions posed in the introduction, as well as to our initial hypothesis. For example, linking mean state changes to changing in variability is what we are doing in the previously mentioned paragraph L438-445. Moving that to the Results, and referring to the question asked, will hopefully increase the level of readability of our Results section.

3. Section 4.2: I had a hard time understanding which dynamical interpretations were from past studies and which were from the figures in this paper. I especially was confused at the "three major aspects" section. For instance, atmospheric heat transport was not discussed elsewhere in this paper but makes an appearance here. I recommend the authors clearly outline where these dynamical interpretations come from their own results and where they are pulling from prior studies. It seems like a lot of this section results from prior studies, which makes me wonder how much of it should be in the introduction.

We agree that our Discussion is currently introducing some literature that was not treated earlier in the paper (for example, in the Introduction). Next to that, how the Discussion is currently written can indeed be confusing as to when we refer to our own results, and when to previously published results. We propose the following:

- We intend to reorder the Introduction (as outlined above), where we will introduce a paragraph on the effects of the different boundary conditions of the Pliocene on the climate. This would include parts of section 4.1 Sensitivity to mid-Pliocene boundary conditions. For example paragraph L417-424 is treating SAT and SIE response to different mid-Pliocene boundary conditions in other climate models, which would actually be relevant to mention in the Introduction.

- We will rephrase some sentences and restructure some parts of the Discussion in such a way that it is clear what is our results or our analysis, and what are findings from previous studies. For example L491-492 ("We observe … Arctic gateways (refs)."); the first part of this sentence is about our results, while the second part refers to findings in previous studies. A proposed rephrasing would be: "Qualitatively the loss of winter sea ice is similar for the Eoi280 and E560 (see Fig 3). However, based on previous research (refs) we know that mid-Pliocene sea-ice loss in the North Pacific is caused by the closed Arctic gateways."

4. Title: I have two thoughts about the title, the second of which relates to Overarching Comment #1. First, I think "warm climate conditions" is too general; the authors only focus on the mid-Pliocene. Second, the title doesn't really seem to capture the main point of the paper. Although the increased dominance of the NPO over the PNA is an interesting result, it seems to me that the main point of this paper is that the mid-Pliocene is a bad analog for future climate change when studying NH winter variability. The authors say this specifically in L13-14 of the abstract.

Thank you, we agree the title can be potentially misleading. The other reviewer also has a comment on the title, so to answer to both, we propose the following title:

"**Mid-Pliocene not analoguous to high CO$_2$ climate when considering Northern Hemisphere winter variability**"

5. Zonality & Azonality: At L185 the concepts of zonality and azonality are introduced and for the rest of the paper I proceeded to get confused about which was which. To begin, I don't think that "azonal" is a common word—for instance, there is no

definition in the AMS glossary. I think this a good first check on whether a term needs to be defined. How do the authors define azonal specifically? Presumably they mean something different than "meridional," correct? How do they decide which mode is zonal and which is azonal? Eyeballing? Second, when I think of the NAO and the NPO, I generally think of them as meridional modes of variability with a center of action in the subtropics and then the other center of action directly above the first in the subpolar region. The authors however say "The NAO is essentially the zonal mode in the NAtl" in L185-186. The PNA, which I think of as a zonal mode with a center of action over the Aleutian Low and one directly eastward over North America, is then identified as the azonal mode. Please explain your reasoning for these descriptions as they seem to be the opposite of my intuition. Last, I found the NPac-z/NPac-a/NAtl-z/NAtl-a terminology confusing, partially because of the confusion around zonal and azonal mentioned before. Why not just say the NPO mode, the PNA mode, the NAO mode, and the EA modes? In the titles of the figures, you say e.g. "NAtl-z (NAO)", so why not just cut out the strange labels and make it so the readers have four less things to keep track of?

Thank you for your comment on the nomenclature used in this paper. It was a choice not to call the NAO the NAO, since we are not sure if the NAO in the midPliocene is the same.

We determine level of zonality by subjectively looking / eyeballing. We have employed a method earlier that used the same definitions used for changing sign of the EOF (L188-190), but this still required a lot of (subjective) tuning so that it worked well. In the end, we are just dealing with two times four EOFs per basin (CR20, E280, Eoi280, E560), so we chose to determine the level of zonality qualitatively.

Furthermore, we propose to stick to the known nomenclature (NAO, PNA, etc) instead of NAtl-z, NPac-a, in order to answer to your remark, as well as a remark from reviewer 1. We agree that the current naming convention, including the definition of 'zonality', might confuse the reader. Sticking to known nomenclature will help, with the side note that the modes in the midPliocene might not be exactly what we expect from the present-day. This is for example also what has been done in previous studies regarding NAO in past climate (e.g. the LGM, see https://journals.ametsoc.org/view/journals/clim/23/11/2010jcli3372.1.xml)

6. Simulation Permutations: It might be informative to show Eoi400-Eoi280 differences. While this isn't precisely the equivalent for comparison with E560-E280 since it's not a doubling, this difference can help us understand whether increased $CO_2$ with mid-Pliocene BCs is similar or different in pattern. This might help with interpretation in the discussion, though it doesn't seem like you have enough simulations to explore the full space of nonlinearity. I think E400 would be needed for that.

Thank you for this suggestion. We have actually performed a 'CO2 doubling' experiment in the mid-Pliocene, the Eoi560 (see Baatsen et al 2022). The results of this simulation can be used to compare to the CO2 doubling with pre-industrial BCs (i.e. Eoi560-Eoi280 versus E560-E280). In addition, it can be used to compare to the mid-Pliocene BC effects at high CO2 (i.e. Eoi560-E560 versus Eoi280-E280). We chose not to include the results of this simulation, as we figured it would not be necessary to answer the research questions.

However, to answer your point as well as reviewer 1, we propose to include some Eoi560 results in the revised manuscript.

We propose to discuss the Eoi400 vs E280 results in more detail in the Discussion, in the section where we currently discuss nonlinearity (L499-506). Specifically, we propose to show a set of figures in the Supplement that we will then refer to, namely the MSLP difference and SLP SD difference for the following five combination (i.e. 5 panels per variable):

1. E560 – E280 (effect of CO2 doubling)
2. Eoi560 – Eoi280 (effect of CO2 doubling with mid-Pliocene BCs)
3. Eoi280 – E280 (effect of mid-Pliocene BCs)
4. Eoi560 – E560 (effect of mid-Pliocene BCs at high CO2)
5. Eoi400 – E280 ('true' mid-Pliocene conditions vs pre-industrial reference)

With this information, we believe that we can properly address nonlinearity, as well as the combined effects of CO2 and other mid-Pliocene BCs in the 'true' mid-Pliocene Eoi400 simulations.

Substantive Comments

1. L19-21: The statement that future climate projections fail to give consistent responses to greenhouse forcing among NH winter variability modes merits a citation.

Later, we cite the IPCC AR6 WGI report to support this claim (L26-27 "in particular because there is no consensus on how the NAM and NAO respond to increasing CO2 concentrations (Eyring et al., 2021)"). We will include that citation in L19-21 as well.

2. L77-87: This paragraph seems out of place. You don't really touch on proxy reconstructions in this paper, just modeled results.

Agreed. We introduced this section to discuss the fact that comparing with proxy reconstructions for this topic would be hard, if not impossible, because of the lack of atmospheric proxies of the midPliocene. But currently, as well as in the newly proposed Introduction structure, the paragraph is not necessary. We propose to move some of the sentences to parts in the Results or Discussion where we interpret and discuss the results.

3. 1.3: I'm confused about what data was previously available and which simulations the authors have run for this study. Some clarification is needed. Have you run these simulations of E560 yourselves? L125 suggests you're using this model specifically because other sensitivity experiments are available.

We understand the confusion. Most simulations had already been performed and have been published in Baatsen et al. (2022). Only the E560 used here is a slight adaptation of the E560 published in Baatsen et al; we treat that in L153-159. To clarify: the published E560

simulation was continued with a minor difference in parametrization that was found to give no influential differences in the E280 case. We will clarify as follows: replace "we perform .." with "we use" or "we employ ..". Also, we will add a sentence stating: "The model specifics and simulations employed in this study have been described in Baatsen et al (2022) and have been run for longer times for use in this study (see spin in Figure S1)."

4. L181 and throughout: NHem is not the universally accepted abbreviation for Northern Hemisphere. I very strongly recommend changing it to the common "NH."

Thank you; we used it to indicate the 1$^{st}$ EOF in the Northern Hemisphere, but since we will adopt the usual terms (either NAM or AO) we can skip using NHem, and will use NH when we mean the Northern Hemisphere.

5. L209: "The mean MSLP difference is very small…" —> are you talking about some area mean value?

Yes, we mean spatial mean MSLP. However, it might not be particularly meaningful (see also answer to reviewer 1). We propose to remove this sentence altogether, and instead compute RMSE (computed per grid point, then averaged).

6. Fig 1, 5, 6: Dashed vs. dash-dotted lines are difficult to discern. I recommend solid for positive, dashed for negative, and thicker line for 0.

We chose to avoid using solid lines in the contour plots, because it was a bit confusing with the coastlines. However, both reviewers have a comment on the readability of the contour lines, so we will reconsider. Your suggestion is appreciated and we will try that.

7. L218: You already defined your acronyms in L181-182. You then redefine them again in L295.

Thank you, it is indeed unnecessary, we will remove here.

8. Fig 2, 3, 4, 6 Captions: Write "minus" or use a minus sign rather than "min" as it currently stands. It's only two more letters and you use "min" at other points to mean minimum.

Thanks for the suggestion, we will use 'minus' in the caption and other places in text where we indicate the difference. (or; write 'difference between x and y')

9. L263: An arctic amplification citation would be useful here.

OK, we propose to cite Serreze & Barry 2011 with a synthesis on Arctic amplification research: https://www.sciencedirect.com/science/article/abs/pii/S0921818111000397

10. L283: "a lot stronger" — this seems arbitrary. I recommend quantifying how much stronger. Similarly, you use "a lot stronger" again in L308 and "a lot weaker" in L513.

We will try to avoid these arbitrary or qualitative descriptions as much as possible. In L283, we propose ".. a lot more obvious ..". In L308 we actually quantify the results already. In L518 we will include a quantification.

11. L284: "significantly" -- do you mean statistically significantly or just substantially weaker?

We mean substantially and will use that word instead (no specific relevance for introducing statistical significance here).

In other cases where we compute the Pearson correlation we will introduce statistical significance (e.g. p-value<0.05).

12. Fig 8: It seems like we're missing some panels here. Unless I'm misreading this figure, you don't show the correlation of jet intensity with NPac-z or jet latitude with NPac-a. Why is that? I recommend adding those panels. Also, it would be worth mentioning somewhere that the Eoi280 scatter in Fig 8d is nonlinear so the linear correlation might not be the best metric.

1. Agreed the 'opposite' scatters and correlations might be interesting, however they are not as relevant / pronounced. We will add this in the supplement and mention the results in the main text with ref to Suppl.
2. We will also add statistical significance here instead of the R^2
3. In L375-377 we mention that the distribution is not unimodal, has two peaks corresponding to the two-jet state in one of the phases. However, we agree it is not explicitly mentioned that the relation might be nonlinear. We will add a sentence "The fact that a split jet exists in the NPO- phase implies that the distribution of jet latitudes is not linear, which suggests that a linear fit might not be the best metric to capture the correlation."

13. L403-405: Citation for split jet and wave breaking?

We propose to include the following relevant studies on wave breaking and modes of winter variability: https://agupubs.onlinelibrary.wiley.com/doi/full/10.1029/2010GL043309 and https://journals.ametsoc.org/view/journals/clim/23/11/2010jcli3372.1.xml

14. L505-506: Different simulation permutations may help with disentangling the effects of BCs vs. CO₂. See Overarching Comment #6.

See our earlier answer on the Eoi560 simulation and nonlinearity.

15. The capitalizations throughout the citations seem to be somewhat arbitrary and using different citation styles. I suggest standardizing to one citation style.

Thank you, we will standardize this.

Minor Typos

-L1: "…we address the question OF whether…" OK, will change this.

-L16-17: "…there is a need to make…" -- avoid the passive voice as able Propose to change into: " … and accurate projections of our future climate is necessary."

-L106: "Next to that…" -- doesn't make sense in this sentence We will remove it

-L114: The community seems to refer to it as "CESM" not "The CESM" when using the acronym You are correct, we will remove "the"

-L117: Why "therefore"? This doesn't make sense to me. "therefore" in the sense that the PlioMIP2 community refers to the this version of CESM as CCSM4 because of its use of CAM4. However, the use of "therefore" is not really necessary and can confuse the reader, so we propose to remove it.

-L201: "It is not be a one-on-one comparison" -- incorrect wording Will remove "be"

-L368: "as well between both azonal modes" -- strange wording Will change this when changing the nomenclature of the modes.

-L381: UK vs. US English spelling is interspersed throughout, e.g., "behavior" here, "behaviour" in L172, 364, 464. I recommend sticking to one or the other. We propose to stick to UK English and will make spelling changes accordingly

-L440: Is STJ subtropical jet? This acronym was not introduced before this line. Yes, it is, and we will introduce it

-L525: "we posed the question OF whether" Will make the proposed change

-L527: Typo-- "analogous" Thank you, will change

-L529: "WHO state" Will change

---

## Author Comment (AC3)

Answers to: EC1

https://doi.org/10.5194/egusphere-2023-757-EC1

Dear editor,

Many thanks for considering our manuscript for publication at WCD and taking the time to act as editor. We acknowledge your critiques and constructive suggestions for improving the manuscript. In this reply we will address your questions and respond to your suggestions.

"This paper presents EOFs of January NH midlatitude circulation variability in the Pliocene from the output of a set of previously published experiments from one model used in the PlioMIP2 project. The authors conclude the Pliocene climate is not an analog for a future climate under increasing CO2 because the variability in NH January is different because of differences in boundary conditions (orography) during the Pliocene vs. modern, **underlining the results from Menemenlis et al. (2021) who showed the Northern Hemisphere stationary wave is greatly reduced in the same model when late-Pliocene boundary conditions are used in place of modern day boundary conditions.**"

We want to make a few clarifying remarks considering this sentence in the first paragraph:
- Although we agree some of our results underline those presented by Menemenlis et al (2021), we treat SLP and jet stream **variability**, while they are not, and mainly considering long-term means.
- The models are indeed very similar (essentially both adaptations of CCSM4), but differ in some important ways: different ocean mixing parameterisations, different coupling between the atm-ocn-ice models, and a different initialisation. Haywood et al (2020) discuss general features of all PlioMIP2 models and it shows that CCSM4-UoT (Menemenlis et al 2021) and CCSM4-Utr (present manuscript & Baatsen et al 2022) have quite different responses in the mean: https://cp.copernicus.org/articles/16/2095/2020/cp-16-2095-2020.pdf

Then, in response to your enumerated items:

"1. The manuscript falls short, however, in providing a dynamical analysis of why the variability and the mean state) changes under Pliocene boundary conditions. Further analysis should be done to demonstrate why the variance in the various patterns changes. For example, why does the variance in the PNA change? Is it due to a reduction in the mean state stationary wave – the main source of energy for the PNA (see, e.g., the discussion on page 237 of Wallace et al. 2023) – but a dynamical analysis should be performed to confirm this. Or is it due to a reduction in ENSO variability? The authors should quantify the different contributions to the change in variance of the PNA. Similarly, evidence through analysis should be provided on why the variance in the NPO changes."

1. **Dynamical analysis:**
   We agree that the dynamical analysis in the current manuscript, aiming to connect changes in variability to changes in the mean state and the boundary conditions, can be improved. We propose to include the following:
   a. Results of a measure that indicates how stationary waves change in the North Pacific. This could be the eddy streamfunction (as in Mememenlis et al), or

Rossby wave source, or another variable that is relevant. This can inform us whether a change in stationary waves is at the cause of changes in SLP variability.

b. Additionally, if our intended analysis at a. does not already provide enough detail, we will include more results on the mean dynamical state of the atmosphere; where we include a measure of velocity potential, (potential) temperature, and/or isentropic density. This can help us to explain whether a change in distribution of mass in the atmosphere is at the cause of changes in the mean jet and jet variability. Both results at a. and b. can be included in the Supplement with reference in the main paper.

c. Changes in ENSO amplitude (e.g. Nino3.4 SD) for all simulations. We know ENSO amplitude changes in the Eoi400 (see f.e. Baatsen et al 2022 or Oldeman et al 2021) but have not included a quantification for Eoi280. We will compute it and include the results, with which we aim to answer why SLP variability is reduced (instead of speculating in the Discussion). A more detailed analysis on ENSO is outside of the scope of this paper, but will be treated in planned future work.

"2. The discussion of why the surface air temperature changes in response to changing boundary conditions is speculative: without a quantitative analysis of the thermodynamic energy budget, one can't discern the relative importance of changes in the mean state stationary wave vs rectified effects of changes in the (PNA) transients. The changes in the mean state circulation would probably create a pattern of warming/cooling in the N. Pacific that is very similar to that in Fig. 3c, but it isn't clear to me that this pattern could result from changes in the variability in the PNA (as is argued in section 4.2). To support this claim, the revised paper should show the rectified effect of changing PNA variance and a quantitative analysis of the relative contributions of the mean state and transient changes to the thermodynamic balance warming tendency (e.g., calculate the changes in $d(\nabla \cdot \overline{v'T'})$, $\delta(\nabla \cdot \overline{v}\,\overline{T})$, $(\nabla \cdot \overline{Q})$, etc, where the overbar denotes time mean and the prime denotes transients)."

2. **SAT changes to boundary conditions**:

a. First, we acknowledge that we cannot discern relative effects of all individual BCs (just gateways effect or just GIS effects) with our simulation suite. Hence, we refer to other studies where more specific sensitivity experiments have been performed.

b. Then, to answer why we see certain surface temperature changes, we will include results of an energy budget analysis. More specifically, we propose to use an energy balance model similar to the one employed in Baatsen et al 2022 and Burton et al 2023. It would essentially be an extension of Fig 11 in Baatsen et al 2022, but then in winter (instead of annual mean) as well as per grid point (instead of zonal mean), so we can discern the contribution in different regions. This will tell us which contribution to the energy balance generates the warming (for example albedo effects or cloud effects), and we can use that information to distinguish the warming processes in E560 and Eoi280.

c. Your suggestion to compute the thermodynamic balance warming tendency is appreciated. However, we are not convinced that such a calculation is necessary to distinguish the SAT changes in the different simulations. Indeed,

it would be interesting and could add valuable information, but we consider this to be outside the scope of the current manuscript.

"3. Concerning the changes in the mean state, Menemenlis et al. (2021) documented that the Northern Hemisphere stationary wave is greatly reduced in this model when late-Pliocene boundary conditions are used in place of modern day boundary conditions. Here, the authors speculate (using the results in section 4.2 of Hurwitz et al) that SLP increases in the Aleutian low in the Pliocene because of increases in SST in the N. Pacific. It is difficult to say for sure (because of the lack of contours and/or the poor resolution in the color bar used in Fig. 3c and other figures) but I don't think the scaling works. Hurwitz et al show a 30 m geopotential response at 850 hPa for a 2C warming in the N. Pacific, which amounts to approximate 3 hPa SLP response (=30 m *(hPa / 8 m) * 850/1000 ) for a 2C anomaly, or 1.5 hPa per 1 C anomaly. In response to Pliocene orography, there is a 16 hPa increase in the Aleutian Low and a ~4 C increase in N. Pacific SST (Figs. 2c and 3c), which is almost three times greater than the response to the prescribed SST anomalies. Indeed, the SST anomalies seem to be a response to the changes in the stationary wave, not the other way around."

3. **Hurwitz et al hypothesis:**
   a. We appreciate your effort at falsifying our proposed hypothesis / speculation. Your rough estimation indeed seems to point out that our comparison with the results by Hurwitz et al is not entirely valid. We propose to remove the section in the Discussion where we make this comparison. Alternatively, we can keep the reference, but mention that the comparison might not be entirely valid, as your estimation shows.
   b. Regarding "lack of contours and/or poor resolution", we will change our continuous colormap to a discrete colormap. In this way, contours will be included, and it will be easier to distinguish specific values.

"4. Another, more likely, cause of the stationary wave response to Pliocene boundary conditions is changes in the tropical Pacific diabatic heating (precipitation). There is a long literature dating back to Simmons et al. (1983) that shows the strength of the Aleutian low and the amplitude of the stationary wave is sensitive to small changes in diabatic heating over the Maritime continent. Figure 1 of Menemenlis et al. (2021) shows that, in response to Pliocene boundary condition, precipitation is reduced over the far western Pacific and increased in the (unrealistic) double ITCZ in central and eastern Pacific. Hence, it would seem changes in the tropical Pacific climatology could easily be responsible for the changes in the climatological mean state Aleutian Low and the stationary wave (at least, in the Pacific), for the weakening and broadening of the climatological mean jet, and for the changes in the variability in the PNA. Simple AMIP experiments using prescribed climatological SSTs taken from the E280 and Eoi280 simulations would illuminate the cause(s) for these changes in the simulations."

4. **Tropical Pacific heating:**
   a. Thank you for this comment. We believe that with including a measure on stationary waves (suggested at 1a.) as well as the energy budget in the Pacific (suggested at 2b.), we will already have some parts available to answer this question. Additionally, we propose to extend our SAT results to lower latitudes, for example 20S, to include the tropics. On top of that, we will include precipitation results in the Supplement, or alternatively as contours on the SAT or SLP results. (Fig 2 or 3).

b.  The suggested AMIP experiments could indeed illuminate the links between tropical heating and the atmospheric response. However, we think that by assessing the tropical Pacific climatology (as outlined in a.) as well as the connection with ENSO (as outlined in 1c.) in the simulations currently treated in the manuscript, we can cover the role of the tropical Pacific in North Pacific variability. Therefore, we do not think that additional AMIP experiments are necessary for answering our research questions, and so we consider them to be outside of the scope of this paper.

c.  To add to the last answer, we have however performed an additional fully coupled experiment, the Eoi560 (in addition to: E280, E560, Eoi280, Eoi400 that are already treated in the current manuscript), which is a 2x pre-industrial $CO_2$ simulation with mid-Pliocene boundary conditions. We share more detail on this simulation at answer 9b.

"5. January and February are special months in the N. Pacific when the jet takes on a more subtropical location and becomes strong and supports less variability – the so-called Pacific mid-winter suppression of the jet. I am not surprised that the EOFs of DJF circulation change in a similar in the Pliocene to those shown in the paper for January (but showing that analysis instead of the analysis of January only would boost the statistical significance of the results). Perhaps even more interesting, it is less clear the other winter months – ONDM, the stormy months in the Pacific – will show the same Pliocene minus modern differences as those in the mid-winter suppression months. Streamlining the introduction and discussion of previous results concerning mean state changes and removing tangential discussion on changes in heat transport in section 4.2 would leave room for a comparison of the changes in variability."

5.  **Variability in (extended) winter months:**
    a.  Thank you for this useful remark. Regarding the increased statistical significance of DJF over January-only results, we propose to repeat the performed analyses for DJF instead of January only. We will include some January-only results, or even extended winter (NDJFM), in the Supplement, to make the comparison between January-only results and DJF results quantitative.
    b.  Streamlining introduction and discussion: We will change the Introduction, as well as better streamline the Discussion, as has been outlined in the answers to Reviewer 1 and 2.
    c.  Ultimately, we think that mid-winter suppression is interesting, but not necessary to treat in order to answer our research questions. Thus, we regard the assessment of Pacific mid-winter suppression of the jet to be outside of the scope of the current manuscript.

"6. Consider analyzing the variability and mean state changes in at least one other climate model used in the PlioMIP2 project. Are your results sensitive to the model used? Fig 1a of the paper shows that the biases in the modern day January stationary wave in the model are large – about twice too large in the N. Atlantic and 40% too large in the N. Pacific – and so too is the variability too large – by a factor of 2 or three."

6.  **Repeat analysis in another model**
    a.  Yes, we expect that the results are sensitive to the model used. This is based on the fact that the PlioMIP2 models show a range of mean state responses (e.g. Haywood et al 2020), different sensitivity to $CO_2$ (e.g. Burton et al 2023),

as well as a range of ENSO responses (Oldeman et al 2021 and Pontes et al 2022). We discuss these results briefly in the Introduction, but we currently do not treat the model dependence in the Discussion. We will include a section in the Discussion where we mention this.

    b. Regarding comparison to reanalysis: the SLP biases are locally large, especially in MSLP and SLP variance (less so in the EOFs). We will explicitly mention the implication of these biases in the Discussion.

    c. Although we can expect differences with different models, we do not think it to be necessary to repeat analyses with another model. Model dependency regarding Pacific variability in the midPliocene will be treated in planned future work.

"7. The use of nonstandard (and apparently arbitrary) assignments of the labels "zonal" and "azonal" terminology to describe well know patterns of atmospheric variability is needlessly confusing. Without further justification, I strongly urge the authors to use standard monikers for these patterns to avoid needlessly confusing the readers. [E.g., the NAO and NPO describe regional-scale patterns of variability featuring meridional dipoles in geopotential, changes in the jet strength, and changes in the meridional location of the storm track. It is difficult to see how that fits with the monikers "zonal" and "azonal".]"

7. **Zonal / azonal terminology**

    a. Thank you for this remark, which echoes the comments of reviewers 1 and 2. So, we will change it accordingly, as we have also mentioned in more detail in the answers to reviewers 1 and 2.

"8. Consider using ERA5 instead of CR20 for the modern "observations", or truncate the CR20 period to start in the early-mid 1900s. The former has 72 years of very good data; the latter is less constrained – especially in the first half of the analysis period used (1836-2015)."

8. **Use ERA5 instead of CR20**

    a. We specifically looked for a reanalysis dataset that would be well suited to compare to our equilibrated pre-industrial simulation results (E280). We considered ERA5, because of its high quality, but decided not to use it since it only spans a short period (relative to our 200 years) and it is a better representation of the present-day (instead of pre-industrial). Instead, we chose to use CR20, exactly for the reason that it covers more of the 'pre-industrial' period (although one can argue that even 1836 is not pre-industrial), and because of the length of the dataset (179 years). Furthermore, the spatial and temporal resolution of the CR20 is still sufficient for comparison to our pre-industrial results. We believe that using CR20 over ERA5 is justified.

"9. I agree with both reviewers that the title doesn't fit the contents of the paper (e.g., the title refers to generic warm climates rather than the late Pliocene) and that adding an analysis of the response to an increase in CO2 under late Pliocene conditions (the change in the pair of experiments Eoi400 and Eoi280) would add new results to the paper (vis à vis the response to increased CO2 under different boundary conditions)."

9. **Title and extra simulation results**

    a. Regarding the title: we agree with you, and both reviewers, that an alternative title would be better. We propose the following title (see also the

reviewer answers): "**Mid-Pliocene not analoguous to high CO$_2$ climate when considering Northern Hemisphere winter variability**"

b. Regarding analysis of response to increased CO$_2$ under Pliocene boundary conditions: we agree with you, and both reviewers, that this would be interesting to include as well as relevant in answering our research question. We have performed a 'CO2 doubling' experiment in the mid-Pliocene, the Eoi560 (see Baatsen et al 2022). The results of this simulation can be used to compare to the CO2 doubling with pre-industrial BCs (i.e. Eoi560-Eoi280 versus E560-E280). In addition, it can be used to compare to the mid-Pliocene BC effects at high CO2 (i.e. Eoi560-E560 versus Eoi280-E280). We chose not to include the results of this simulation in the present manuscript, as we figured it would not be necessary to answer the research questions. We propose to include some Eoi560 results in the revised manuscript. See also more detail on this in the answers to reviewers 1 and 2.

---

## Author Response (AR1)

Author's response

A point-by-point response to the reviews can be found on EGU sphere Discussion:
https://egusphere.copernicus.org/preprints/2023/egusphere-2023-757/

We provided a revised manuscript and revised supplement. Both files have been uploaded, including a track-changes file (latexdiff). Below we will highlight the main changes:

- The title of the manuscript has been changed,
- The order of the paragraphs in the Introduction has been changed, and some context previously in the Discussion was moved to the Introduction,
- The mean winter climate results have been updated to include mean precipitation response, and previous figures 2-4 have been merged into two figures.
- The modes of variability have been renamed (i.e. NPac-z is now just NPO),
- The results include a new subsection where the link between tropical convection and North Pacific jet and SLP variability is explained (includes some new results and dynamical explanation),
- The Discussion has been changed such that it is clearer what is interpretation based on the results presented in the manuscript, and what is discussion related to literature,
- Some interpretation of the results has been removed from the Discussion, and replaced by a discussion based on the new results in section 3.3.3,
- The discussion includes a new subsection briefly discussing the results of a simulation with mid-Pliocene boundary conditions and 560ppm $CO_2$, related to the additivity of responses to the mid-Pliocene boundary conditions and to elevated $CO_2$,
- The discussion includes a section on model performance in light of the reanalysis and compared to the PlioMIP2 ensemble,
- All figures have been changed to increase visibility and readability,
- The supplementary material has been updated with some additional results.

We believe that this revised manuscript is an improvement of the previous version. We hope that the reviewers and editor will consider the revised manuscript for publication in WCD.

---

## Referee Report (RR1)

Second Review of: Mid-Pliocene not analogous to high CO2 climate when considering Northern Hemisphere winter variability (Oldeman et al., 2023)
Authors: Arthur M. Oldeman, Michiel L.J. Baatsen, Anna S. von der Heydt, Aarnout J. van Delden, & Henk A. Dijkstra

This is my second review of this study. I congratulate the authors on strengthening their manuscript. I found the overall study had a much more coherent storyline, the figures were easier to read, and I thought the added tropical Pacific discussion was interesting. I have only a few minor comments remaining before this study is accepted for publication.

Energy budget: In your response to the Editor you mentioned you would be completing an energy budget analysis to respond to his concern about why SAT changes in response to changing BCs. I either missed this, in which case it needs to be made more obvious, or it wasn't included. Either way, I think this concern needs to be addressed more clearly.
L321 & L333: I didn't follow why you think the changes in precipitation around Greenland are related to sea ice.
Figure 3: This isn't major, but I suggest flipping the color axis on the contours so that blue is wetter and red is dryer for panels b and c. I think it's then a bit more logical to quickly read the plots.
Figure 6: I had a hard time differentiating between bold and not bold font in these squares. I suggest using underlines or asterisks to show the $p<0.005$ values instead.

Typos:
L203: Should be a colon rather than a semicolon.
L349-350: "a lot more" and "a bit less" are both qualitative phrasings. You have statistical significance here, I suggest sticking to the quantitative descriptors.
L370: "differences are that" is awkward grammar.
L378: "the northern node is retreated polewards" is awkward. Maybe it should be "has retreated" instead.
L394-396: I got confused with your use of parentheses here and missed the message you were trying to make. This isn't a case of "the mode is positively (negatively) correlated with..." sort of use you use slightly later in the manuscript. Please revisit this sentence.
L400-401: Similar strange parenthetical structure as comment above.
L476: "a jet stream weak in strength" should be "a weak jet"?
L478: Similarly, "a jet less variable in strength" and "a jet more variable in latitude" is strange phrasing.
L483: "indexc"
L485: "WEP is established before" should be "WEP has been established"
L498: "summarizing 2. - 4b." -- what are you referring to? Figures?
Figure 9 Caption: "correlation coefficient in the caption" -- I think you mean in the legend.
L510: "that shows" rather than "that show" since the subject of that sentence is the "study."
L577 & L281: You've referred to it as "Supplementary" material more throughout, so I suggest changing "Supplement" here.
L654: I feel like there should be a "can" or "should" between "climate" and "be" in this sentence.
L679: "might not be" is rather weak language compared to the rest of your conclusion which says the Mid-Pliocene should not be used as an analogue.

L683: "we think that it might" can just be "it may"

Supplementary Material S6: I believe the second Figure S10 at the end of the section should be Figure S11.

---

## Editor Decision (ED1)

Second Review of: Mid-Pliocene not analogous to high CO2 climate when considering Northern Hemisphere winter variability (Oldeman et al., 2023)
Authors: Arthur M. Oldeman, Michiel L.J. Baatsen, Anna S. von der Heydt, Aarnout J. van Delden, & Henk A. Dijkstra

This is my second review of this study. I congratulate the authors on strengthening their manuscript. I found the overall study had a much more coherent storyline, the figures were easier to read, and I thought the added tropical Pacific discussion was interesting. I have only a few minor comments remaining before this study is accepted for publication.

Energy budget: In your response to the Editor you mentioned you would be completing an energy budget analysis to respond to his concern about why SAT changes in response to changing BCs. I either missed this, in which case it needs to be made more obvious, or it wasn't included. Either way, I think this concern needs to be addressed more clearly.
L321 & L333: I didn't follow why you think the changes in precipitation around Greenland are related to sea ice.
Figure 3: This isn't major, but I suggest flipping the color axis on the contours so that blue is wetter and red is dryer for panels b and c. I think it's then a bit more logical to quickly read the plots.
Figure 6: I had a hard time differentiating between bold and not bold font in these squares. I suggest using underlines or asterisks to show the $p<0.005$ values instead.

Typos:
L203: Should be a colon rather than a semicolon.
L349-350: "a lot more" and "a bit less" are both qualitative phrasings. You have statistical significance here, I suggest sticking to the quantitative descriptors.
L370: "differences are that" is awkward grammar.
L378: "the northern node is retreated polewards" is awkward. Maybe it should be "has retreated" instead.
L394-396: I got confused with your use of parentheses here and missed the message you were trying to make. This isn't a case of "the mode is positively (negatively) correlated with..." sort of use you use slightly later in the manuscript. Please revisit this sentence.
L400-401: Similar strange parenthetical structure as comment above.
L476: "a jet stream weak in strength" should be "a weak jet"?
L478: Similarly, "a jet less variable in strength" and "a jet more variable in latitude" is strange phrasing.
L483: "indexc"
L485: "WEP is established before" should be "WEP has been established"
L498: "summarizing 2. - 4b." -- what are you referring to? Figures?
Figure 9 Caption: "correlation coefficient in the caption" -- I think you mean in the legend.
L510: "that shows" rather than "that show" since the subject of that sentence is the "study."
L577 & L281: You've referred to it as "Supplementary" material more throughout, so I suggest changing "Supplement" here.
L654: I feel like there should be a "can" or "should" between "climate" and "be" in this sentence.
L679: "might not be" is rather weak language compared to the rest of your conclusion which says the Mid-Pliocene should not be used as an analogue.

L683: "we think that it might" can just be "it may"

Supplementary Material S6: I believe the second Figure S10 at the end of the section should be Figure S11.

Reviewer #2: comments on the revised manuscript

This study is much improved from the initial submission, and the authors have done an excellent job at addressing my concerns, including substantial new analysis, a section dedicated to the likely role of tropical SSTs, and substantial restructuring of the paper. The new title now accurately represents the findings. I am happy to recommend publication, subject to the following minor suggestions:

Line 17. How is the response of the climate to increased CO2 determined by natural variations?

Line 59. 'lowering of the Rocky mountains' implies to me that the mountains reduced in height over time, rather than, as I think you mean, that the mountains were lowered in the model simulations of the Pliocene.

Line 130. I know you are focussed on the atmosphere, but given your results on the impacts of the tropical convection, which is almost certainly related to tropical SSTs, I think it is useful to also given information on the ocean model and resolution.

Line 261: "The weak but distinct eastern node with opposite sign in the CR20 disappears in the E280" – from my interpretation of figure 1g, the eastern opposite sign node is present in the shading, but not the black/white contours, which from the caption means it is present in E280 but not the CR20, the opposite to the text.

I recommend being slightly more clear that Eoi280 is mid-pliocene boundary conditions, not mid-pliocene conditions, as stated in, for example, line 293.

Fig. 6. I understand that you don't need the AO column in panels b and c, but then you also don't need the top row – it's a little less clear to me why you would make the different panels a slightly different shape rather than just retain the triangles for the auto-correlations.

Line 455. Suggest to add ('blue line') to help readers.

Section 3.3.3
Point 1a. Suggest '…Walker circulation, and lead to a northward shift…' for clarity that the northward shift isn't being reduced.
Point 3. It isn't clear to me why reduced Rossby wave forcing from the tropics would lead to a North Pacific jet that is more variable in latitude – is this from the literature or from your results?

Careful with correlations vs causality here – I'm not sure you can conclusively say that the change in Rossby wave tropical forcing causes changes to the jets, which cause the changes in the NPO/PNA patterns instead of changes in the Rossby wave tropical forcing causing changes in the NPO/PNA patterns which lead to change in the jet, can you? The zonal wind changes in Fig 9b and c look like they are likely approximately geostrophic if the pressure changes are equivalent barotropic.

Line 503. 'thus lead to'

Line 505. I like that you have included the RWS analysis, but I think it is worth mentioning in the main manuscript that you include RWS analysis in the supplementary material, rather than just "more extensive analysis"

Line 513: dates for citations shouldn't be in parentheses within parentheses

Line 533: in E560 the warming is 'mostly' due to greenhouse gases? Is there any other forcing in this simulation?

Line 621. Can you clarify that those 3 changes seen by 'most' PlioMIP2 models (and thus, I assume, in the multi-model mean), are also seen in the CCSM4-Utr model?

Line 631. How does orography adjust as a feedback in response to climate change?

Line 654. Grammar: … we address the question of whether the mid-Pliocene climate can be…

Supplementary: S5.3. I'm unsure how the RWS can be a wave guide for the jet stream. We typically think of the jet stream as waveguides for waves from a RWS.

---

## Author Response (AR2)

Second Review of: Mid-Pliocene not analogous to high CO2 climate when considering Northern Hemisphere winter variability (Oldeman et al., 2023)
Authors: Arthur M. Oldeman, Michiel L.J. Baatsen, Anna S. von der Heydt, Aarnout J. van Delden, & Henk A. Dijkstra

This is my second review of this study. I congratulate the authors on strengthening their manuscript. I found the overall study had a much more coherent storyline, the figures were easier to read, and I thought the added tropical Pacific discussion was interesting. I have only a few minor comments remaining before this study is accepted for publication.

Energy budget: In your response to the Editor you mentioned you would be completing an energy budget analysis to respond to his concern about why SAT changes in response to changing BCs. I either missed this, in which case it needs to be made more obvious, or it wasn't included. Either way, I think this concern needs to be addressed more clearly.

- We will include a brief energy budget analysis, using a simple energy balance model following Baatsen et al (2022), Hill et al (2014) and Heinemann et al (2009). We will include methods and results in the Supplementary material and mention the conclusions briefly in the main manuscript.
- Using this method, we can separate the surface warming between two simulations in contributions related to planetary albedo (i.e. shortwave fluxes), effective emissivity (i.e. longwave fluxes) and meridional heat transport. We can split the emissivity contribution in warming related to longwave cloud forcing, and warming related to clearsky emissivity, e.g. from greenhouse gases or lapse rate feedbacks.
- In section 4.1.2 'The role of mean surface temperature response', we will replace the first paragraph with conclusions from the energy balance model, with reference to the Supplement. We propose the following: "Qualitatively, the SAT response to increased CO2 and to the mid-Pliocene BCs is similar. **However, an energy budget analysis reveals that the radiative forcings leading to the warming are not the same (shown in Supplementary material Figure S10). The surface warming in the E560 is explained by changes in effective emissivity from the increased greenhouse gas concentrations. The surface temperature response in the Eoi280 is a combination of changes in planetary albedo, as well as effective emissivity. The changes in emissivity here are related to lapse rate feedbacks and changes in water vapour. The warming due to changes in planetary albedo is mostly related to changes in vegetation and lakes. Albedo changes from the reduced Greenland ice sheet do not contribute to warming in January because the incoming solar radiation is at a minimum. This is different in the annual mean, as seen in Baatsen et al (2022). In both simulations, warming in areas with sea-ice loss is related to reduced emissivity from increased evaporation leading to increased cloud cover."**

Hill et al (2014): https://cp.copernicus.org/articles/10/79/2014/
Heinemann et al (2009): https://doi.org/10.5194/cp-5-785-2009

L321 & L333: I didn't follow why you think the changes in precipitation around Greenland are related to sea ice.

- A decrease in sea-ice leads to increased precipitation through increased local evaporation and convection, see e.g.
https://www.pnas.org/doi/abs/10.1073/pnas.1504633113

- We will make the following change in the manuscript: "…and a concentrated precipitation increase is seen South of Greenland. **The precipitation increase in the Arctic is related to a** retreat of sea-ice in that area (not shown) **through increased local evaporation (Kopec et al 2015)**, agreeing with the local SAT increase and MSLP decrease."

Figure 3: This isn't major, but I suggest flipping the color axis on the contours so that blue is wetter and red is dryer for panels b and c. I think it's then a bit more logical to quickly read the plots.
- Agreed, we will make that change accordingly

Figure 6: I had a hard time differentiating between bold and not bold font in these squares. I suggest using underlines or asterisks to show the p<0.005 values instead.
- We will use asterisks instead

Typos:
L203: Should be a colon rather than a semicolon. Will change
L349-350: "a lot more" and "a bit less" are both qualitative phrasings. You have statistical significance here, I suggest sticking to the quantitative descriptors. We will remove 'a lot' and 'a bit' altogether since the sentence already includes information that these differences are statistically significant
L370: "differences are that" is awkward grammar. We will change: "The spatial pattern is very similar, **with notable difference being that …**"
L378: "the northern node is retreated polewards" is awkward. Maybe it should be "has retreated" instead. We will change to: "while the northern node is **weakened and shifted polewards.**"
L394-396: I got confused with your use of parentheses here and missed the message you were trying to make. This isn't a case of "the mode is positively (negatively) correlated with..." sort of use you use slightly later in the manuscript. Please revisit this sentence. We will change to: "… the **correlation between the AO and EA is stronger than the correlation between the AO and NAO**, which is not very obvious based on the similarities of the spatial patterns **of these modes**."
L400-401: Similar strange parenthetical structure as comment above. We will change as follows: "The AO shows the strongest correlation with the EA **(PNA) in the NA (NP)**, and simultaneously the EA and PNA show a stronger correlation **compared to the E$^{280}$.**"
L476: "a jet stream weak in strength" should be "a weak jet"? We will change into 'a weak jet' as suggested
L478: Similarly, "a jet less variable in strength" and "a jet more variable in latitude" is strange phrasing. In L474, we propose to include ".. jet stream that is less variable in strength **(in terms of zonal wind speeds, Figure 7c)**…" so that it is clarified what is meant by strength. In L478 we will add ".. more variable in latit**udinal location …**" as in L475 before.
L483: "indexc" Thanks, we will change to 'index'
L485: "WEP is established before" should be "WEP has been established" Agreed, we will change accordingly
L498: "summarizing 2. - 4b." -- what are you referring to? Figures? We are referring to the steps of the mechanism as elucidated before in the section. We will add "summarizing **steps 2. – 4b. of the mechanism**"
Figure 9 Caption: "correlation coefficient in the caption" -- I think you mean in the legend. Indeed, we will change accordingly

L510: "that shows" rather than "that show" since the subject of that sentence is the "study."
Thanks, we will change this
L577 & L281: You've referred to it as "Supplementary" material more throughout, so I suggest changing "Supplement" here. We will consistently use "Supplementary material" throughout
L654: I feel like there should be a "can" or "should" between "climate" and "be" in this sentence. We will add 'can', as in the research question posed in the Introduction
L679: "might not be" is rather weak language compared to the rest of your conclusion which says the Mid-Pliocene should not be used as an analogue. Agreed, we will change to "is not", which is more consistent also with the title and abstract
L683: "we think that it might" can just be "it may" We will change accordingly
Supplementary Material S6: I believe the second Figure S10 at the end of the section should be Figure S11. Indeed, we will change this accordingly.

Reviewer #2: comments on the revised manuscript

This study is much improved from the initial submission, and the authors have done an excellent job at addressing my concerns, including substantial new analysis, a section dedicated to the likely role of tropical SSTs, and substantial restructuring of the paper. The new title now accurately represents the findings. I am happy to recommend publication, subject to the following minor suggestions:

Line 17. How is the response of the climate to increased CO2 determined by natural variations? We meant to say that natural variability is a part of the response of the climate to increased CO2, not the other way around. We will change it: "… the response of the climate system itself**, including feedbacks and natural variability,** to increased CO2."

Line 59. 'lowering of the Rocky mountains' implies to me that the mountains reduced in height over time, rather than, as I think you mean, that the mountains were lowered in the model simulations of the Pliocene. Indeed, we mean that the mountains had been lowered in the simulations. We will change this to: "… primarily attributed to the **reduced height of the Rocky Mountains in the model simulations.**"

Line 130. I know you are focussed on the atmosphere, but given your results on the impacts of the tropical convection, which is almost certainly related to tropical SSTs, I think it is useful to also given information on the ocean model and resolution. Agreed. We will include "… The model version used here employs **the ocean model POP2 and** the atmosphere module CAM4, …" and in L135: "The atmospheric grid … levels**, while the ocean grid has a nominal 1° (1.25° x 0.9°) horizontal resolution.**"

Line 261: "The weak but distinct eastern node with opposite sign in the CR20 disappears in the E280" – from my interpretation of figure 1g, the eastern opposite sign node is present in the shading, but not the black/white contours, which from the caption means it is present in E280 but not the CR20, the opposite to the text. We acknowledge that this is confusing. We will change this sentence: "**The center of gravity of the strong negative node shifts eastward in the E$^{280}$, with regards to the CR20,** likely because the variance shifts more eastward in the E$^{280}$." This is slightly different information, but we think this is actually more relevant for the total context.

I recommend being slightly more clear that Eoi280 is mid-pliocene boundary conditions, not mid-pliocene conditions, as stated in, for example, line 293. Agreed, we will make sure to consistently refer to the Eoi280 simulation as 'mid-Pliocene boundary conditions'.

Fig. 6. I understand that you don't need the AO column in panels b and c, but then you also don't need the top row – it's a little less clear to me why you would make the different panels a slightly different shape rather than just retain the triangles for the auto-correlations. The motivation to remove one column was to reduce figure width (apart from the column not adding any extra information). To resolve this, we can add the extra column so that the shape is more consistent, and we will slightly increase the fontsize.

Line 455. Suggest to add ('blue line') to help readers. We will add this.

Section 3.3.3
Point 1a. Suggest '…Walker circulation, and lead to a northward shift…' for clarity that the northward shift isn't being reduced. We propose to change the order of this sentence to the following, for clarity: "Shifts in the tropical mean state, **specifically the … of the ITCZ,** reduce mean precipitation in the west-equatorial Pacific (WEP)."

Point 3. It isn't clear to me why reduced Rossby wave forcing from the tropics would lead to a North Pacific jet that is more variable in latitude – is this from the literature or from your

results? This conclusion is from both our results and in support of literature. However, in the current phrasing it is indeed unclear and suggestive. We propose to change to the following, where we include an extra reference in support of the mechanism:

"**3a.** Since the jet stream acts a wave guide for Rossby waves (e.g. Branstator, 2002), reduced upper-tropospheric Rossby wave activity over East Asia leads to a North Pacific jet stream that is **weaker (Figure 2c) and** less variable in strength (Figure 7c).

**3b. A weaker jet stream is associated with Rossby wave breaking downstream (e.g. Woollings 2010, de Vries et al. 2013) which leads to a jet** more variable in latitudinal location **(e.g. Rivière 2010) over the central and eastern North Pacific** (Figure 7d)."

where Woollings (2010) and Rivière (2010) have already been cited in the manuscript, and de Vries et al. (2013) would be added: https://link.springer.com/article/10.1007/s00382-013-1699-7

Careful with correlations vs causality here – I'm not sure you can conclusively say that the change in Rossby wave tropical forcing causes changes to the jets, which cause the changes in the NPO/PNA patterns instead of changes in the Rossby wave tropical forcing causing changes in the NPO/PNA patterns which lead to change in the jet, can you? The zonal wind changes in Fig 9b and c look like they are likely approximately geostrophic if the pressure changes are equivalent barotropic. This is a good point. However, we do not think any changes in the text are needed. Based on our results, one can indeed not discern whether changes in the jet lead to changes to modes of variability, or the other way around, since we only show correlation coefficients. In the present manuscript, we already tried to avoid causality between jet and PNA/NPO by using words like 'correlates with' and 'implies', instead of 'causes'.

Line 503. 'thus lead to' Thanks, will change

Line 505. I like that you have included the RWS analysis, but I think it is worth mentioning in the main manuscript that you include RWS analysis in the supplementary material, rather than just "more extensive analysis" We will change this into: "**More analysis** on the connection between tropical convection and Rossby wave activity, **including a computation of the Rossby wave source**, can be found in …". In addition, we will include a sentence in the Methods section (section 2.2): "**Additionally, we performed a calculation of the Rossby wave source, which is included in the Supplementary material.**"

Line 513: dates for citations shouldn't be in parentheses within parentheses. Thank, we will make this change

Line 533: in E560 the warming is 'mostly' due to greenhouse gases? Is there any other forcing in this simulation? This is a mistake, we will remove the word 'mostly'

Line 621. Can you clarify that those 3 changes seen by 'most' PlioMIP2 models (and thus, I assume, in the multi-model mean), are also seen in the CCSM4-Utr model? Yes, we will add: "Most PlioMIP2 models, **including CCSM4-Utr**, show …"

Line 631. How does orography adjust as a feedback in response to climate change? Here we mean that adjustment to orography changes is a feedback, but since this is unclear we will change it: "… such as adjustment to changes in ice sheets **and to changes in orography,** …"

Line 654. Grammar: … we address the question of whether the mid-Pliocene climate can be… We will change this.

Supplementary: S5.3. I'm unsure how the RWS can be a wave guide for the jet stream. We typically think of the jet stream as waveguides for waves from a RWS. This is a mistake and should be the other way around (as is correctly stated in the manuscript L473). We will change this accordingly.

---

## Author Response (AR3)

**Authors reply on coeditor comment 26 january 2024**

Dear Dr. Oldeman,
Thanks for your constructive and helpful modifications in response to the reviewers. I have only one minor request before accepting the paper (which should take less than one hour) which concerns the discussion of the processes responsible for the surface temperature changes. The equation you use to do this (from Heinemann et al 2009) doesn't balance energy; e.g., the turbulent heat fluxes are not accounted for, nor are changes in the petitioning of heat transport divergence in the atmosphere and ocean which would affect surface temperature without directly changing the emissivity of the atmosphere. Rather, it is best described as "a model for diagnosing surface temperature assuming a gray atmosphere without turbulent heat fluxes." Nonetheless, Heinemann et al shows that it works fairly well to reproduced changes in temperature simulate by a climate model -- so I am fine with you applying it here. However, I would like to see a few words in the main text about the equation applied (perhaps including a phrase like the one in quotes above).
David

- We agree that the 'model' used is a tool to diagnose surface temperatures based on a simple energy budget analysis (ie shortwave and longwave radiation following from the climate model), but doesn't really balance energy. Taking your useful feedback, we propose to add the following sentence in the paragraph:
- "We diagnose surface temperatures with shortwave and longwave radiation, using a simple model assuming a gray atmosphere without turbulent heat fluxes (following Heinemann et al. (2009) and Baatsen et al. (2022), results shown in Supplementary material Figure S10)."

PS. In lines 546-7 of the revised text, did you mean an increase in atmospheric emissivity due to increases in cloud fraction (and perhaps water vapor)? A decrease in atmospheric emissivity would cool the surface by allowing more greater fraction of the longwave emitted from the surface to reach the top of the atmosphere. I think the confusion here is because of the non-standard way emissivity is defined in Heinemann et al.

- You are correct, to remove any confusion and to be consistent with the phrasing in the rest of the paragraph, we will change this into: "… is related to **changes in** emissivity from increased evaporation …"